# High Resolution Model Intercomparison Project (HighResMIP v1.0) for CMIP6

R.J. Haarsma[1], M. Roberts[2], P. L. Vidale[3], C. A. Senior[2], A. Bellucci[4], Q. Bao[5], P. Chang[6], S. Corti[7], N. S. Fučkar[8], V. Guemas[8], J. von Hardenberg[7], W. Hazeleger[1,9,10], C. Kodama[11], T. Koenigk[12], L. R. Leung[13], J. Lu[13], J.-J. Luo[14], J. Mao[15], M. S. Mizielinski[2], R. Mizuta[16], P. Nobre[17], M. Satoh[18], E. Scoccimarro[4], T. Semmler[19], J. Small[20], J.-S. von Storch[21]

[1]Royal Netherlands Meteorological Institute, De Bilt, The Netherlands
[2]Met Office Hadley Centre, Exeter, UK
[3]University of Reading, Reading, UK
[4] Centro Euro-Mediterraneo per i Cambiamenti Climatici, Bologna, Italy
[5]LASG, Institute of Atmospheric Physics, Chinese Academy of Sciences, Beijing , China P.R.
[6]Texas A&M University, College Station, Texas, USA
[7]National Research Council – Institute of Atmospheric Sciences and Climate, Italy
[8]Barcelona Super Computer Center, Earth Sciences Department, Barcelona, Spain
[9]Netherlands eScience Center, Amsterdam, The Netherlands
[10]Wageningen University, The Netherlands
[11]Japan Agency for Marine-Earth Science and Technology, Japan
[12]Swedish Meteorological and Hydrological Institute, Norrköping, Sweden
[13]Pacific Northwest National Laboratory, Richland, USA
[14]Bureau of Meteorology, Australia
[15]Environmental Sciences Division and Climate Change Science Institute, Oak Ridge National Laboratory, Oak Ridge, TN, USA
[16]Meteorological Research Institute, Tsukuba, Japan
[17]Instituto Nacional de Pesquias Espaciais, Brazil
[18]AORI University of Tokyo, Tokyo, Japan
[19]Alfred Wegner Institute, Bremerhaven, Germany
[20]Nacional Center for Atmospheric Research, Boulder, Colorado, USA
[21]Max-Planck-Institute for Meteorology, Hamburg, Germany

Correspondence to: R.J. Haarsma (haarsma@knmi.nl)

**Abstract**
Robust projections and predictions of climate variability and change, particularly at regional scales, rely on the driving processes being represented with fidelity in model simulations. The role of enhanced horizontal resolution in improved process representation in all components of the climate 40 system is of growing interest, particularly as some recent simulations suggest the possibility for significant changes in both large-scale aspects of circulation, as well as improvements in small-scale processes and extremes.
However, such high resolution global simulations at climate time scales, with resolutions of at least 50 km in the atmosphere and 0.25° in the ocean, have been performed at relatively few research 45 centers and generally without overall coordination, primarily due to their computational cost. Assessing the robustness of the response of simulated climate to model resolution requires a large multi-model ensemble using a coordinated set of experiments. The Coupled Model Intercomparison Project 6 (CMIP6) is the ideal framework within which to conduct such a study, due to the strong link

to models being developed for the CMIP DECK experiments and other model intercomparison projects (MIPs).

Increases in High Performance Computing (HPC) resources, as well as the revised experimental design for CMIP6, now enables a detailed investigation of the impact of increased resolution up to synoptic weather scales on the simulated mean climate and its variability.

The High Resolution Model Intercomparison Project (HighResMIP) presented in this paper applies, for the first time, a multi-model approach to the systematic investigation of the impact of horizontal resolution. A coordinated set of experiments has been designed to assess both a standard and an enhanced horizontal resolution simulation in the atmosphere and ocean. The set of HighResMIP experiments is divided into three tiers consisting of atmosphere-only and coupled runs and spanning the period 1950-2050, with the possibility to extend to 2100, together with some additional targeted experiments. This paper describes the experimental set-up of HighResMIP, the analysis plan, the connection with the other CMIP6 endorsed MIPs, as well as the DECK and CMIP6 historical simulation. HighResMIP thereby focuses on one of the CMIP6 broad questions: "what are the origins and consequences of systematic model biases?", but we also discuss how it addresses the World Climate Research Program (WCRP) grand challenges.

**1 Introduction**

Recent studies with global high resolution climate models have demonstrated the added value of enhanced horizontal atmospheric resolution compared to the output from models in the CMIP3 and CMIP5 archive. They showed significant improvement in the simulation of aspects of the large scale circulation such as El Niño Southern Oscillation (ENSO) (Shaffrey et al., 2009; Masson et al., 2012), Tropical Instability Waves (Roberts et al., 2009), the Gulf Stream (Kirtman et al., 2012) and Kuroshio (Ma et al., 2016) and their influence on the atmosphere (Minobe et al., 2008; Chassignet and Marshall, 2008; Kuwano-Yoshida et al., 2010; Small et al., 2014, Ma et al., 2015), the global water cycle (Demory et al., 2014), snow cover (Kapnick and Delworth 2013), Atlantic inter tropical convergence zone (ITCZ) (Doi et al., 2012),  jet stream (Lu et al., 2015; Sakaguchi et al., 2015), storm tracks (Hodges et al., 2011) and Euro-Atlantic blocking (Jung et al., 2012). High horizontal resolution in the atmosphere has a positive impact in representing the non-Gaussian probability distribution associated with the climatology of quasi-persistent low frequency variability weather regimes (Dawson et al., 2012).  In addition, the increased resolution enables a more realistic simulation of small scale phenomena with potentially severe impacts such as tropical cyclones (Shaevitz et al., 2015; Zhao et al., 2009; Bengtsson et al., 2007; Murakami et al., 2015; Walsh et al., 2012; Ohfuchi et al., 2004; Bell et al., 2013; Strachan et al., 2013, Walsh et al., 2015), tropical-extratropical interactions (Baatsen et al., 2015; Haarsma et al., 2013) and polar lows (Zappa et al., 2014). Other phenomena that are sensitive to increasing resolution are ocean mixing, sea-ice dynamics, diurnal precipitation cycle (Sato et al., 2009; Birch et al., 2014; Vellinga et al., 2016), quasi bienennial oscillation (QBO) (Hertwig et al., 2015), the Madden-Julian oscillation (MJO) representation (Peatman et al., 2015), atmospheric low-level coastal jets and their impact on sea surface temperature (SST) bias along eastern boundary upwelling regions (Patricola and Chang, 2016, Zuidema et al., 2016) and monsoons (Sperber et al., 1994; Lal et al., 1997; Martin, 1999). The improved simulation of climate also results in better representation of extreme events such as heat waves, droughts (Van Haren et al., 2015) and floods. Enhanced horizontal resolution in ocean models can also have beneficial impacts on the simulations. Such impacts include improved simulation of boundary currents, Indonesian Throughflow and water exchange through narrow straits, coastal currents such as the Kuroshio, Leeuwin Current, and  Eastern Australian Current, upwelling, oceanic eddies, SST fronts (Sakamoto et al., 2012; Delworth et al.,, 2012; Small et al., 2015), ENSO (Masumoto et al., 2004;  Smith et al., 2000; Rackow et al., 2016) and sea ice drift and deformation (Zhang et al., 1999; Gent et al., 2010). Although enhanced resolution in atmosphere and ocean models had a beneficial impact on a wide range of modes of internal variability, the relative short high resolution simulations make it difficult to sort that out in detail due to large decadal fluctuations in interannual variability in for instance ENSO (Sterl et al., 2007).

The requirement for a multitude of multi-centennial simulations, due to the slow adjustment times in the Earth system, and the inclusion of Earth System processes and feedbacks, such as those that involve biogeochemistry, has meant that model resolution within CMIP has progressed relatively slowly. In CMIP3, the horizontal typical resolution was 250 km in the atmosphere and 1.5° in the ocean, while more than seven years later in CMIP5 this had only increased to 150 km and 1° respectively. Higher resolution simulations, with resolutions of at least 50 km in the atmosphere and 0.25° in the ocean, have only been performed at a relatively few research centers until now, and generally these have been individual "simulation campaigns" rather than large multi-model comparisons (e.g. Shaffrey et al., 2009; Navarra et al., 2010; Delworth et al., 2012, Kinter et al., 2013; Mizielinski et al., 2014; Davini et al., 2016). Due to the large computer resources needed for these simulations, synergy will be gained if they are carried out in a coordinated way, enabling the construction of a multi-model ensemble (since ensemble size for each model will be limited) with common integration periods, forcing and boundary conditions. The CMIP3 and CMIP5 databases provide outstanding examples of the success of this approach. The multi-model mean has often proven to be superior to individual models in seasonal (Hagedorn et al., 2005) and decadal

forecasting (Bellucci et al., 2015) as well as in climate projections (Tebaldi and Knutti, 2007) in response to radiative forcing. Moreover, significant scientific understanding has been gained from analyzing the inter-model spread and attempting to attribute this spread to model formulation (Sanderson et al., 2015).

The primary goal of HighResMIP is to determine the robust benefits of increased horizontal model resolution based on multi-model ensemble simulations – to make this practical vertical resolution will not be considered. The argument for this is that the scaling between horizontal and vertical resolution must obey N/f, where N is the Brunt-Väisälä frequency and f the coriolis parameter. This implies a factor of 100, between horizontal and vertical resolution, which is well satisfied by the model configurations in the HighResMIP group. In addition components such as aerosols will be simplified to improve comparability between models. The top priority CMIP6 broad question for HighResMIP is "what are the origins and consequences of systematic model biases", which will focus on understanding model error (applied to mean state and variability), via process-level assessment, rather than on climate sensitivity. This has motivated our choices in terms of proposed simulations, which emphasize sampling the recent past and the next few decades where internal climate variability is a more important factor than climate sensitivity to changes in greenhouses gases (Hawkins and Sutton, 2011), at least at regional scales.

The use of process-based assessment is crucial to HighResMIP, since we aim to better understand the dynamical and physical reasons for differences in model results induced by resolution change, in order to increase our trust in the fidelity of models. Such process understanding will either contribute to bolster our confidence in results from lower resolution (but with greater complexity) CMIP simulations, or to enable a better understanding of the limitations of such models. There are an increasing number of studies suggesting that, in individual models, important processes are better represented at higher resolution indicating ways to potentially increase our confidence in climate projections (e.g. Vellinga et al., 2016). A wide variety of processes will be assessed, from global and regional drivers of climate variability, down to mesoscale eddies in atmosphere and ocean – in the atmosphere these include tropical cyclones (Zhao et al., 2009; Bell et al. 2013; Rathmann et al., 2014; Roberts et al., 2015; Walsh et al., 2015) and eddy-mean flow interactions (Novak et al., 2015; Schiemann et al., 2016), while for the ocean they are an important mechanism for mesoscale air-sea interactions (Chelton and Xie, 2010; Bryan et al., 2010; Frenger et al., 2013; Ma et al., 2015; 2016), trans-basin heat transport (e.g. Agulhas leakage) (Sein et al., 2016), convection and oceanic fronts.

HighResMIP will coordinate the efforts in the high-resolution modeling community. Joint analysis, based on process-based assessment and seeking to attribute model performance to emerging physical climate processes (without the complications of (bio)geochemical Earth System feedbacks) and sensitivity of model physics to model resolution, will further highlight the impact of enhanced horizontal resolution on the simulated climate. As the widespread impact of horizontal resolution, in the range of a few hundred  to about ten kilometer, on climate simulation has been demonstrated in the past, it is expected that HighResMIP will contribute to many of the grand challenges of the WCRP, and hence such analysis may begin to reveal at what resolution in this range particular processes can be robustly represented.



The remainder of this manuscript is structured as follows. Section 2 gives an overview of the simulations, while section 3 describes the tiers of simulation in detail. Section 4 makes links between these and the CMIP6 DECK and other CMIP6 MIPs, section 5 describes the data storage and sharing plans, and section 6 and 7 describe the analysis and potential application plans. Conclusions and discussion are contained in Section 8.  Several appendices contain more detail of the experimental design and forcing.


## 2 Outline of HighResMIP simulations

The main experiments will be divided into Tiers 1, 2 and 3. They are illustrated in Fig. 1. We provide an outline of these different tiers, with more detail in section 3. Each set of simulations comprise model resolutions at both a standard and a high resolution, where the standard resolution model is expected to be used in a set of CMIP6 DECK simulations and are considered the entry card for HighResMIP,.

The Tier 1 experiments will be historical forced atmosphere (ForcedAtmos) runs for the period 1950-2014. A number of centers have already performed similar high resolution simulations and published their results (CAM5 Bacmeister et al., 2014; HadGEM3 Mizielinski et al., 2014; NICAM Satoh et al., 2014; EC-Earth Haarsma et al., 2013) hence these runs should not present prohibitively large technical difficulties. Restricting the ForcedAtmos runs to the historical period also makes it possible for numerical weather prediction (NWP) centers to contribute to the multi-model ensemble. Nineteen international groups have expressed interest in completing these simulations as shown in Appendix 9.1. All centers participating in HighResMIP are obliged to participate at least in Tier 1.

The coupled experiments in Tier 2 are more challenging, but provide an opportunity to understand the role of natural variability, due to the centennial scale, and to investigate the impact of high resolution on future climate. Although a few centers have previously carried out high resolution coupled simulations such as SINTEX-F2, GFDL, Hadley, MIROC and CESM (Masson et al., 2012; Delworth et al., 2012; Mecking et al., 2016; Sakamoto et al., 2012; Small, 2014), considerable issues including mean-state biases, climate drift and ocean spin-up remain. Due to these issues and the large amount of computer resources needed to complete both a reference and a transient simulation, fewer centers (currently six) are confirmed participants for these experiments. The period of the coupled simulations is 1950-2050.

Future atmosphere-only simulations for the period 2015-2100 will be carried out in in Tier 3. Although the future period covers the entire present century, the simulations can for computational reasons be restricted to mid-century (2050).

For a clean evaluation of the impact of horizontal resolution, additional tuning of the high resolution version of the model should be avoided. The experimental set-up and design of the standard resolution experiments will be exactly the same as for the high-resolution runs. This enables the use of HighResMIP simulations for sensitivity studies investigating the impact of resolution. If unacceptably large physical biases emerge in the high resolution simulations, all necessary alterations should be thoroughly documented. The requirement of no additional tuning is more relevant for the coupled runs because atmosphere alone models are strongly constrained by the prescribed SSTs.

### 2.1 Common Forcing fields

To focus on the impact of resolution in the design of the HighResMIP simulations should minimize the difference in forcings and model set-up that would hamper the interpretation of the results. Most of the forcing fields are the same as those used in the CMIP6 Historical Simulation that are described separately in this Special Issue (Eyring et al., 2016) and are provided via the CMIP6 data portal. For the future time period, GHG and aerosol concentrations from a high-end emission scenario of the Shared Socioeconomic Pathways (SSP) will be prescribed, which in the following will be denoted by SSPx. A summary of the differences in forcing between CMIP6 AMIPII protocol and the Tier 1 and Tier 2 simulations is given in Table I.

### 2.1.1 Aerosol

A potential large source of uncertainty is the aerosol forcing – for the same aerosol emissions, different models can simulate very different aerosol concentrations, hence producing different
radiative forcing. In HighResMIP, each model will use its own aerosol concentration background climatology. To this will added an anthropogenic time-varying, albeit uniform, forcing provided via the MACv2-SP method by Stevens et al. (2015). These will be computed using a new approach to prescribe aerosols in terms of optical properties and fractional change in cloud droplet effective radius to provide a more consistent representation of aerosol forcing. This will provide an aerosol
forcing field that minimizes the differences between models as well as reducing the need for model tuning. This method is also the standard method in CMIP6 DECK for models without interactive aerosols.

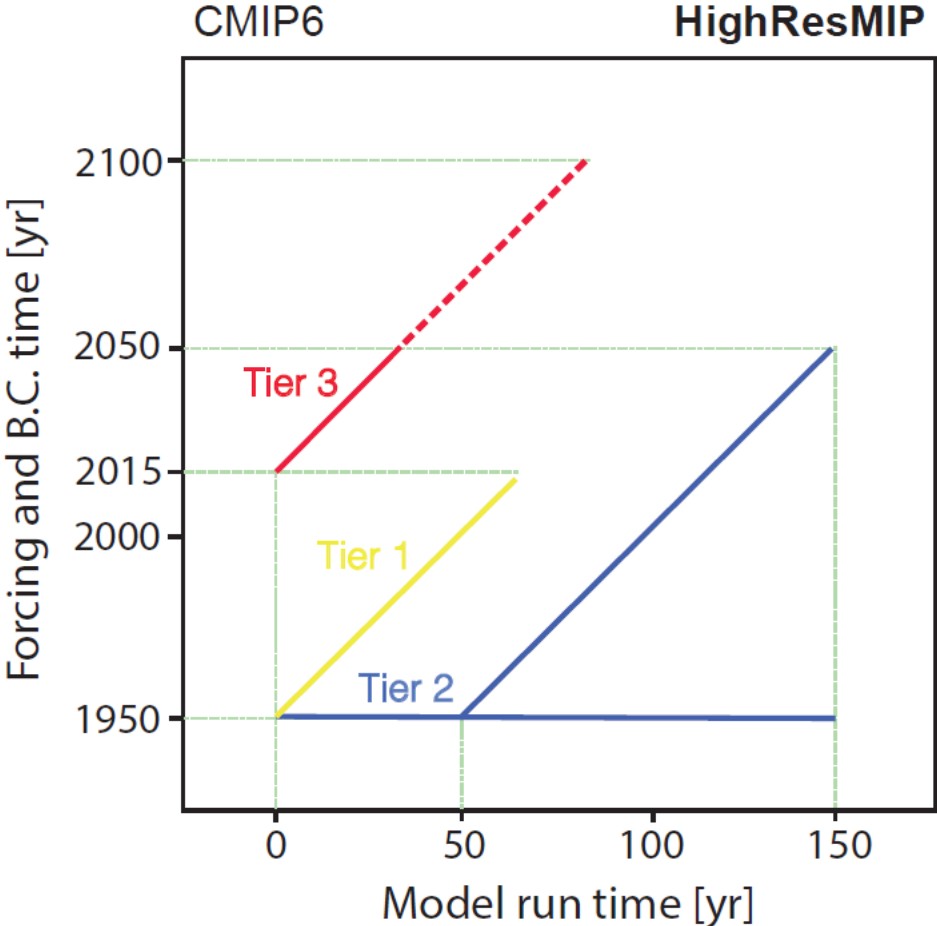

**Figure 1:** Schematic outline of the Tiers 1, 2 and 3. Tier 1 is a 64 year AMIP simulation from 1950-2014 with historical forcings. The first part of Tier 2 (coupled ocean-atmosphere simulations) consists of a 50 year integration starting from 1950 initial state under 1950's conditions. Thereafter this simulation will be continued by two branches of 100 years: one continuing with the 1950's forcing (control run) and the other using until 2014 historical forcings and for 2015-2050 SSPx
(scenario run). Tier 3 is the extension of Tier 1 from to 2014 to2050 (obliged, solid line) and 2051-2100 (optional, dashed line) for SSPx.

### 2.1.2 Land surface

The land surface properties will also be different from the CMIP6 AMIPII protocol. Given the
requirement to make model forcing as simple as possible to aid comparability, the land-surface

properties will be climatological seasonally varying conditions of leaf-area index (LAI), with no dynamic vegetation and a constant land-use/land-cover consistent with the present day period, centered around 2000. Consideration was given to attempting to further constrain land surface properties to be more similar between groups, but this was rejected given the complex and different ways in which remotely-sensed values are mapped to model land surface properties. However, an additional targeted experiment has been included to further investigate the sensitivity to land surface representation. This is outlined in section 9.3 "Targeted additional experiments".

### 2.1.3 Initialization and spin-up of atmosphere-land system
As discussed in Eyring et al. (2016), the initialization of land surface and atmosphere require several years of spin-up to reach quasi-equilibrium before the simulation proper can begin. We recommend this is done using the first few years of the forcing datasets before restarting in 1950. We further recommend that the initial condition for the atmosphere and land for 1950 (for the highresSST-present, and the highres-1950 experiment) come from the ERA-20C reanalysis from January 1950. If this is not possible then the exact procedure used should be fully documented by each group.

### 3 Detailed description of Tiers
### 3.1 Tier 1 simulations: ForcedAtmos runs 1950-2014 – *highresSST-present*
The target for high resolution is 25-50 km, which is significantly higher than the typical CMIP5 resolution of 150 km. These ForcedAtmos runs will also be performed with the standard resolution that is used for the DECK and historical simulation.

The 1950-2014 simulation period is longer than the DECK AMIPII that spans 1979-2014. This is motivated primarily by work in many groups interested in climate variability over multi-decadal timescales, focusing on different phases of climate modes of variability such as Atlantic meridional oscillation (AMO) and Pacific decadal oscillation (PDO), as well as improved sampling of ENSO teleconnections (Sterl et al., 2007). The longer period will also improve the robustness of assessing the difference in variability between standard and higher resolution simulations, as well as being important for statistics of teleconnections (e.g. Rowell, 2013). Furthermore, the longer period of integration will enable a much more robust assessment of the ability of models to simulate known modes and their phases of variability, which is a big issue for climate risk assessment and decadal predictions where the combined effect of the global warming signal and natural variability will be considered.

The recommended ensemble size for the high resolution simulations is three, but due to their computational cost many centers will probably be able to simulate only one member. Therefore although an ensemble size of three is recommended it is not a requirement to participate in HighResMIP. The small ensemble size or absence of it will be insufficient for a rigorous estimate of the contribution of the internal variability to the total climate signal. However, by using a strictly common protocol in the various participating centers, the effective multi-model ensemble size will be much larger, enabling a much wider sampling than –pre-HighResMIP of the multi-model robustness of resolution impacts. In addition, if models can be proven to be portable, the ensemble size could be increased if auxiliary computer resources should become available at a later stage.

### 3.1.1 SST and sea-ice forcing
Although there is a significant forcing of the ocean by the atmosphere, in particular in the midlatitudes (Wu et al., 2007), many recent studies have shown that gradients in SST associated with fronts and ocean eddies can have significant influence on the atmosphere via changes in air-sea fluxes (Minobe et al., 2008; Parfitt et al., 2015; Ma et al., 2015; O'Reilly et al., 2015). Similarly, there is evidence that daily variability rather than monthly smoothed forcing can influence model

simulations (de Boisseson et al., 2012; Woollings et al., 2010). Since the high resolution simulations will approach 25km, this means there is a requirement for a daily, $\frac{1}{4}$ degree dataset for a period longer than satellite-based datasets (such as Reynolds et al., 2002) are able to provide. Hence, we will use a new dataset based on HadISST2 (Rayner et al., 2016, Kennedy et al., 2016) which has these properties for both SST and sea-ice concentration for the period 1950-2014 – in addition, it provides an ensemble of historic realizations which can potentially be used to produce multiple ensemble members. It should be noted that the use of a daily, 1/4 degree data set will also have adverse effects. This is an inevitable consequence of AMIP runs. In these runs the ocean has an infinite heat capacity, with a detoriative impact on the phase relationships between SSTs, overlying atmosphere and surface fluxes (Barsugli and Battisti, 1998; Sutton and Mathieu, 2003). Although beneficial for the processes discussed above, the daily, 1/4 degree data will be for instance less optimal for the simulation of extremes over land (Cassou, 2015) and MJO's (DeMott et al., 2015).

**3.2 Tier 2 simulations: Coupled runs**

The coupled simulations are also aimed at addressing questions of model bias in both mean state and variability similar to the ForcedAtmos simulations. There are many examples from previous studies (e.g. Scaife et al., 2011; Bellucci et al., 2010) where these biases become much more evident in the coupled context compared to the forced atmosphere simulations. The systematic comparison between uncoupled (Tier 1) and coupled simulations for the 1950-2050 period, under different horizontal resolutions, will stimulate novel process-oriented studies tackling the origins of well-known biases affecting climate models, such as the double-ITCZ tropical bias.

**3.2.1 Control  - *control-1950***

These coupled runs will be the HighResMIP equivalent of the pre-industrial control, here being a 1950's control using fixed 1950s forcing. The forcing consists of GHG gases, including $O_3$ and aerosol loading for a 1950s (~10 year mean) climatology.

This will allow an evaluation of the model drift. The initial ocean conditions are taken from version 4 of the Met Office Hadley Centre "EN" series of data sets of global quality controlled ocean temperature and salinity profiles and monthly objective analyses (EN4  Good et al., 2013) over an average period of 1950-1954. As described below, a short spin-up with these forcings is required (~50 years) to produce initial conditions for both the 100 year simulation within this control, as well as for the historic simulation described in 3.2.2.

**3.2.2 Historic – *hist-1950***

These are coupled historic runs for the period 1950-2014 using an initial condition taken from 3.2.1. For this period the external forcings are the same as in Tier 1 (see Table 1).

**3.2.3 Future – *highres-future***

These are the coupled scenario simulations 2015-2050, effectively a continuation of the 3.3.2 historic simulation into the future. For the future period the forcing fields will be based on CMIP6 SSPx. Other forcings are detailed in Table 2.

The atmospheric component of the coupled models will be the same as in the Tier 1 simulations. The minimum resolution for the high resolution ocean model is 0.25°. This enables the ocean to resolve some mesoscale variability (compared to non-eddy permitting models), particularly in the tropics, which has been shown to change the strength of atmosphere-ocean interactions (Kirtman et al., 2012). It also aligns the resolution of the ocean with that of the atmosphere – the ideal atmosphere/ocean resolution ratio remains an open scientific question.

The period of the historic coupled integrations is chosen to be the same as in the Tier 1 simulations. The future end-date is based on a compromise between what is computationally affordable by a sufficient number of centers (~100 years of integration) and what is scientifically relevant.

We again emphasize our interest in model error (bias, fidelity in representation of climate processes and variability) rather than climate sensitivity or transient climate response in configuring these coupled simulations, in particular whether any changes in process representation have an influence on patterns of climate variability and change. As discussed before the number of ensemble members that will be possible, at least initially, in HighResMIP will not be sufficient to fully address internal

variability, but it will form an important baseline set of simulations from which already preliminary robust conclusions can be extracted, and should be useful for many of the other CMIP6 MIPs (e.g. DCCP, GMMIP, CORDEX, CFMIP).

The HighResMIP simulations will enable the simulation of weather systems with short time scales

that can provoke strong air-sea interactions such as tropical cyclones. Hence, high frequency coupling between ocean and atmosphere is required: a 3hr or 1hr frequency is highly recommended so that the diurnal time scale can be resolved, assuming sufficient vertical model resolution in the upper ocean.

**3.2.4 Spin-up**
Due to the large computer resources needed, a long spin-up to (near) complete equilibrium is not possible at high resolution (and hence for consistency will not be used at standard resolution). We recommend an alternative approach which will use the EN4 (Good et al., 2013) analyzed ocean state representative of 1950 as the initial condition for temperature and salinity. To reduce the large

initial drift a spin-up of ~50 years will be made using constant 1950s forcing. Thereafter, the control run continues for another 100 years with the same forcing and the scenario run for the 1950-2050 period is started (Fig. 1). The difference between control and scenario can then be used to remove the continuing drift from the analysis. Output from the initial 50 years spin-up should be saved to enable analysis of multi-model drift and bias, something that was not possible in previous CMIP

exercises, with the potential to better understand the processes causing drift in different models.

**3.3 Tier 3 simulations: ForcedAtmos runs 2015-2050 (2100) –** *highresSST-future*
The Tier 3 simulations are an extension of the Tier 1 atmosphere-only simulations to 2050, with an option to continue to 2100. To allow comparison with the coupled integrations the same scenario

forcing as for Tier 2 (SSPx) will be used. However, since all the HighResMIP models will have the same SST and sea-ice forcing (described below), comparison of the Tier 2 and Tier 3 simulations can help to disentangle the impact of a model bias from forced response. This could be useful for applications such as time of emergence (e.g. Hawkins and Sutton, 2012). The forcing fields and scenario are shown in Table 2.


**3.3.1 Detailed description of SST and Sea-Ice forcing**
The future SST and sea-ice forcing is detailed in Appendix 9.2. It broadly follows the methodology of Mizuta et al. (2008), enabling a smooth, continuous transition from the present day into the future. The rate of future warming is derived from an ensemble mean of CMIP5 RCP8.5 simulations, while

the interannual variability is derived from the historic 1950-2014 period. Using SST derived from CMIP5 RCP8.5 in conjunction with a CMIP6 SSPx GHG forcing introduces an inconsistency. However, given the wide range of climate sensitivity among the climate models and the small differences in the model response up to 2050 for different scenario's we argue that this inconsistency is minor.


### 3.4 Further targeted experiments

In addition to the Tier 1-3 simulations above, discussions with other CMIP6 MIP participants have suggested several targeted experiments that would enable further investigation of specific processes and forcings, as well as potentially informing future CMIP protocols. These are optional experiments, and as such the details of the experimental design will be described in Appendix 9.3. In brief they comprise:

a) Leaf Area Index (LAI) experiment – *highresSST-LAI*
   Impact of using a common LAI dataset in models, linking with LS3MIP
b) Impact of SST variability on large scale atmospheric circulation – *highresSST-smoothed*
   Impact of using a smoothed SST and sea-ice forcing dataset, linking with OMIP
c) Idealized forcing experiments with CFMIP – *highresSST-p4K, highresSST-4co2*
   CFMIP-style experiments to investigate impact of model resolution
d) Abrupt 4xCO2 increase in coupled climate model – *highres-4CO2*
   CFMIP-style experiment to investigate role of ocean resolution on ocean heat uptake
e) Tier 2 and 3 using RCP8.5 instead of SSPx - *highres-RCP85*
   For centers that need to start their simulations before the availability of SSPx

### 4 Connection with DECK and CMIP6 endorsed MIPs
#### 4.1 DECK

For the high resolution models, completing the full set of CMIP6 DECK simulations is too expensive in terms of computer resources. Hence, there is an assumption that groups participating in HighResMIP will complete a set of DECK simulations with the standard resolution model. The HighResMIP simulations will in that case be considered as sensitivity experiments with respect to the standard resolution DECK runs, that are the entry cards for HighResMIP. If groups are not able to do this, because for instance the only available configuration is with prescribed SSTs, which is often the case for NWP centers, they can still participate in HighResMIP but their simulations will only be visible as HighResMIP and not as CMIP6 runs.

Although there will be no DECK simulations at the high resolution, the comparisons between the standard resolution simulations within HighResMIP and the DECK simulations will be informative in themselves. The relevance of HighResMIP is that the significant step in horizontal resolution enables us to clarify some of the outstanding climate science questions remaining from CMIP3 and CMIP5 exercises.

For the Tier 1 simulations, there is a strong link with the CMIP6 AMIPII simulations – the latter are likely to provide multiple ensemble members per modeling center, but using slightly different boundary conditions and forcings (SST, sea-ice, aerosols, LAI and land use). Hence this comparison will be informative of the impacts of these changes at the standard resolution common to both AMIPII and HighResMIP. In addition, the multiple ensemble members will provide a measure of internal variability, to assess whether the high resolution simulation lies outside this envelope.

#### 4.2 CMIP6 endorsed MIPs

HighResMIP, as one of the CMIP6 endorsed MIPs, has links with a number of other MIPs. Collaboration with those will enhance the synergy.

*GMMIP* for global monsoons

There is well-known sensitivity of monsoon flow and rainfall to model resolution in the West African monsoon, Indian monsoon and possibly East Asian monsoon. As stated in GMMIP the monsoon rainbands are usually at a maximum width of 200 km. Climate models with low or moderate

resolutions are generally unable to realistically reproduce the mean state and variability of monsoon precipitation for the right reasons. This is partly due to the model resolution. The Tier 1 ForcedAtmos runs of HighResMIP will be used in Task-4 of GMMIP to examine the performance of high-resolution models in reproducing both the mean state and year-to-year variability of global

monsoons. As tropical monsoonal rainfall is sensitive to small scale topography, high resolution has potential to improve this. On the other hand, there is strong evidence of the importance of coupled ocean-atmosphere interactions for the summer monsoon dynamics (Robertson and Mechoso, 2000; Robertson et al., 2003; Wang et al., 2005; Nobre et al., 2012). Consideration was given to starting the HighResMIP from 1870 to better compare with GMMIP, but it would not be affordable for many

groups. In addition, the quality of observational and reanalysis datasets during the earlier period, to assess the modelled variability and processes, is questionable.

*RFMIP*
HighResMIP intends to use the MACv2.0-SP simplified aerosol forcing being partly produced and
analyzed in RFMIP (Stevens et al., 2015). Additionally, assessment of its impact at different resolutions will contribute to understanding this simplified forcing. The impact of different aerosols on atmospheric circulation and teleconnections in the coupled climate system has been shown before and is likely dependent on model resolution (e.g. Chuwah et al., 2016).

*CORDEX*
CORDEX regional downscaling experiments provide focused downscaling over particular regions. Comparison between these and global HighResMIP simulations can give insight into the relative importance of global scale teleconnections and interactions, against enhanced local resolution and local processes. HighResMIP can (potentially) provide boundary conditions for downscaling and
provide a stimulus to cloud resolving simulations, but data volumes are likely to prohibitive so this will be left to individual groups to coordinate.

*OMIP for ocean analysis and initial state*
There is potential to jointly examine the spin-up issues in both forced ocean (OMIP) and coupled
(HighResMIP) simulations, to improve the understanding of how we might better initialize coupled climate or forced ocean simulations and minimize initialization shock and the required integration time. The targeted experiment 9.3.2 to understand the impact of mesoscale SST variability is another joint area of research. We will also exchange diagnostic/analysis techniques to understand ocean circulation changes at different resolutions.

*LS3MIP*
Within the scope of LS3MIP on understanding the land-atmosphere interactions at different horizontal resolutions, HighResMIP can provide useful datasets to evaluate the role of soil moisture on extreme events, as well as the impact of LAI forcing datasets on model variability and mean state
at different resolutions via targeted experiment 9.3.1.

*DynVAR*
Increase of horizontal resolution may also improve the stratospheric basic state through vertical propagation of small-scale gravity waves, which in turn may affect tropospheric circulation. The
sensitivity of such troposphere-stratosphere dynamical interactions on horizontal resolution will be analyzed by the DynVAR community, and HighResMIP has actively coordinated with the DynVAR diagnostic request to make this possible.


*CFMIP*

Targeted experiments in 9.3.3, to look at the clouds and feedback response in different resolution models, can be assessed in conjunction with CFMIP experiments using the standard resolution model.

*SIMIP*

Coordination of sea ice diagnostic request with SIMIP will enable coordinated assessment of the impact of model resolution on sea ice conditions and processes. Indeed, sea ice drift, deformation and leads (Zhang et al., 1999; Gent et al., 2010) have been shown to be highly sensitive to model resolution in single-model studies. The robustness of these conclusions should be assessed in a

coordinated multi-model exercise such as HighResMIP.

**5 Data storage and sharing**

The storage and distribution of high resolution model data is a challenging issue. Since the resolution of HighResMIP approaches the scales necessary for realistic simulation of synoptic weather

phenomena, daily and sub-daily data will be stored to allow the investigation of weather phenomena such as those related to midlatitude storms, blocking, hurricanes and monsoon systems. However, high-frequency output of all 3-dimensional fields will not be affordable to store. Careful considerations are needed to limit the high-frequency output. The considerations should take into account that the information relevant for the end users is concentrated at or near the land surface

where people live so that it is desirable to store surface and near-surface variables in high temporal and spatial resolution. Furthermore, in order to evaluate the HighResMIP-ensemble, the high-frequency output should contain variables for which high-frequency observations are available as well.

HighResMIP output data will conform to all the CMIP requirements for standardization. The CMIP6 data and diagnostic plan (Juckes et al., 2016) describes the diagnostic request for all the CMIP6 MIPs. This data request, including that of HighResMIP is available from the CMIP6 website. The data and diagnostic plan will be finalized during the boreal summer of 2016. An estimate of the amount of data that needs to be stored is given at http://clipc-services.ceda.ac.uk/dreq/tab01_3_3.html.

The data storage is divided in three priorities. This is based on a balance between the HighResMIP data request (**{** HYPERLINK "http://clipc-services.ceda.ac.uk/dreq/u/HighResMIP.html" **}**) to answer scientific questions and the large data volumes involved. Priority 1 should be possible for all centers. Priority 2 and 3, involve large data volumes and more specific questions. HighResMIP groups commit

to archiving at least the priority 1 data request diagnostics on an Earth System Grid Federation (ESGF) node. The very large data volumes mean that it may be difficult to transfer all of priority 2 and 3 data, and hence a different methodology is needed to cope with this. Discussions with other international data centers are planned to further enable collaborative analysis. In the European Horizon 2020 project PRIMAVERA, the JASMIN platform (STFC/CEDA, UK) will be used for data

exchange and as a common analysis platform. In future, it would be a more efficient management of global resources to move analysis tools to data storage centers. The European Copernicus Climate Data Store may also provide useful future avenues for data storage and sharing, which will be explored. Further, the project will explore a close collaboration with the European EUDAT initiative (http://www.eudat.eu), which is developing data storage, preservation, staging and sharing services

suitable for extremely large datasets.

One useful approach may be to provide spatially and/or temporally coarsened model output on the ESGF, which would enable initial analysis compared to DECK simulations, and indicate which avenues of analysis may require full model resolution output, with manageable remaining volumes. It would

then also be available for any automated assessment tools on the ESGF.

**6 Analysis plan**

The analysis will focus on the impact of increasing resolution on the simulation of different climate phenomena that are strongly biased in coarse resolution models and that could potentially benefit from higher resolution. The robustness of the impact of increasing resolution on the simulation of weather and climate phenomena such as extreme weather events, atmospheric eddy – jet stream interactions, atmospheric blocking events, typical ocean model biases and ocean model drift among the different HighResMIP models will be investigated and their response to global warming assessed as well as their interannual variabilities.

The increased resolution will permit evaluating whether horizontal resolution alone can generate a better simulation of regional climates. The analysis will therefore also have a focus on regional climate and relative teleconnections. Because HighResMIP will enable a more detailed simulation of small-scale weather systems, the scale interaction between these systems and the large scale circulation will be another focus of the analysis plan. The benefit of atmosphere-ocean coupling at these high resolutions will be addressed as well since we can compare the AMIP-style simulations with fully coupled simulations. Not all modeling centers may be able to afford eddy-resolving ocean simulations; nevertheless, where possible, it will be interesting to investigate scale interactions in the ocean as well.

**Five initial foci for analyses have been identified:**

**6.1 Regional climates**

Current climate risks assessments rely on output from ensembles of relatively coarse resolution global climate models or on their downscaled products (e.g. CORDEX) in addition to observations. For Europe, around 15 regional modelling groups downscaled ERA-interim simulations at 50 km and 12.5 km resolution (http://www.euro-cordex.net). Furthermore, historical and future simulations of about 10 different CMIP5 models have been downscaled by a similar number of regional climate models. Also for other regional domains as e.g. Africa (Klutse et al., 2015), North America (Mearns et al., 2013) or the Arctic (Koenigk et al., 2015), multi-model downscaling simulations have been performed. While the regional models generally fail to improve the large scale atmospheric circulation, probably due to inconsistencies at their lateral boundaries and insufficient vertical resolution, they show added value in the representation of precipitation, in complex terrain and of meso-scale phenomena as e.g. polar lows (Rummukainen, 2015).

A recent study by Jacob et al. (2014) showed that the high-resolution Euro-CORDEX-simulations provide a more realistic representation of precipitation extremes over Europe and a larger increase of extreme precipitation in future simulations compared to the global models. Generally, the regional CORDEX-simulations show a more sensitive response of precipitation to changes in greenhouse gas concentrations compared to their driving global models. However, the bias in the lateral boundary conditions from coarse resolution climate models can strongly affect the simulations in the regional models, such as shown for precipitation trends over Europe by van Haren et al. (2014, 2015).

The HighResMIP simulations will provide the first ensemble of global models with a comparable resolution to the current generation regional models. This will allow for a direct comparison of user-relevant parameters in HighResMIP to the CORDEX results. The comparison will focus on statistics and physics of meteorological events such as intense rainfall, droughts, storms and heat waves. A comparison of the simulation of extreme events in the global models (which are self-contained and include global small-scale to large-scale interactions) and in regional models (forced at the boundary by another model, and typically a one-way downscaling) will be made. Results from various studies (e.g. Scaife et al., 2011; Kirtman et al., 2012), analyzing the benefits of high resolution in the ocean in

one single global model, indicate that increased resolution in global models leads to an improved simulation of large scale phenomena such as the North Atlantic Current system and related surface temperature gradients. The impact of such improvements on blocking and storm tracks and the downstream effect on European climate variability and extremes will be analyzed and compared to CORDEX-results. Comparing HighResMIP results, with a globally high resolution, to results from both standard resolution global models and regional CORDEX simulations with a locally high resolution domain (but boundaries based on coarse resolution CMIP5 models) will give us insights into the importance of realistic large scale climate conditions for local climate variations and extremes.

Studying internal variability and long-term change of the Northern Hemisphere sea ice cover in the coupled HighResMIP simulations will enable us to explore the impact of better resolved sea ice dynamics on Arctic and global climate. Preliminary tests conducted at 1/4 and 1/12° with the NEMO-LIM3 ocean-sea ice model indicate not only stable results, but also realistic heterogeneities and intermittency behaviors in the sea ice cover. HighResMIP will be the perfect testbed to assess whether these increases in resolution have to be conducted in conjunction with development in model physics (rheology in this case), or if the two can be done separately. Differences between perennial 1950 and historical simulation will further our understanding of Arctic warming amplification and long-term future of sea ice cover superimposed with pronounced natural variability, using methods outlined by Fučkar et al. (2015).

## 6.2 Scale interactions

The improved simulation of synoptic scale systems in HighResMIP enables us to analyze multi-scale phenomena such as large-scale circulation, tropical and extratropical cyclones, MJO, tropical wave, convection and cloud in a seamless manner. For example, tropical cyclogenesis has known links to multi-scale phenomena including monsoon, synoptic-scale disturbances, and MJO (e.g. Yoshida and Ishikawa, 2013). Even for the dynamical storm-track, which may be thought satisfactorily resolved by low-resolution climate models, its bias in latitudinal position is related to the cloud amount bias in CMIP5 models (Grise and Polvani, 2014). Existing high-resolution atmosphere simulations suggest that the characteristics of the jet stream (Hodges et al., 2011) and blocking (Jung et al., 2012) will be improved by higher resolution. The MJO, and diurnal precipitation cycle are also of great interest. Such analysis, requiring high frequency data, has implications for the output diagnostics – see Section 5 and Juckes et al. (2016).

In addition, the role of air-sea interactions at the mesoscale, such as analyzed by Chelton and Xie (2010), Bryan et al. (2010) and Ma et al. (2015), can be assessed across models to understand the impact of resolution and the potential feedbacks in the system that may change the mean state.

Regarding the ocean, multi-scale phenomena can be discussed in a similar way. By resolving eddies and having a lower dissipation due to refined resolution, the cold bias in the northwest corner, the pathway of the Gulf stream / North Atlantic current, the Southern Ocean warm bias as well as the Agulhas current have been shown to be substantially improved (Sein et al., 2016). Even at an intermediate ¼° resolution which is not eddy-resolving, improvements have been shown (Marzocchi et al., 2015). This has strong links with OMIP.

## 6.3 Process studies

Process-level assessment of the simulated climate will give us some insights to improve the physics scheme in the climate models at a range of resolutions. Satellite simulators will be applied to the HighResMIP model output to evaluate cloud and precipitation processes in detail (e.g., Hashino et al., 2013). After the launch of the EarthCare satellite (planned in 2018; Illingworth et al., 2015), a

new dataset including vertical distribution of cloud, precipitation, and vertical velocity is expected to be available. The fact that the horizontal resolution of the climate model is approaching that of the satellite observations also motivates us to accelerate synergetic studies between models and observations.

Process studies will aim to pin down the reasons for potentially better capturing small-scale and consequently large-scale phenomena with increasing resolution. Such process understanding will be the basis for developing schemes or error correction methods that could potentially compensate for not capturing a range of processes in standard resolution models.

This topic has links with RFMIP (aerosols), LS3MIP (land surface processes), CFMIP (clouds), SMIP (sea-ice) and DynVar (troposphere-stratosphere processes).

## 6.4 Extremes and hydrological cycle

Many aspects of climate extremes are associated with the hydrological cycle, together with dynamical drivers such as mid-latitude storm tracks and jets. Analysis following Demory et al. (2014) will assess the multi-model sensitivity of the global hydrological cycle to model resolution, and convergence of moisture over land and ocean. In the tropics, the hydrological extremes due to monsoon systems and interactions between land and atmosphere (Vellinga et al., 2016; Martin and Thorncroft et al., 2015) will be investigated in conjunction with GMMIP. On a regional scale the extremes and hydrological cycle will be analyzed in collaboration with CORDEX. For extremes associated with surface processes there are links with LS3MIP.

In mid-latitudes, the representation of storm tracks and jet streams will be assessed. Novak et al. (2015) investigated the role of meridional eddy heat flux on the tilt of the North Atlantic eddy-driven jet. This behavior may partly explain the dominant equatorward bias of the jet stream in generations of global climate simulations with model resolutions much coarser than 50km (Kidston and Gerber, 2010; Barnes and Polvani, 2013; Lu et al., 2015). Biases in the jet stream position have been found to correlate with the meridional shift of the jet position in a warmer climate (Kidston and Gerber, 2010).

Atmospheric rivers play a key role in the global and regional water cycle (Zhu and Newell, 1998; Ralph et al., 2006; Leung and Qian, 2009; Neiman et al., 2011; Lavers and Villarini, 2013), and hydrological extremes, and have been shown to be sensitive to model resolution (Hagos et al., 2015). In both North Pacific and North Atlantic, uncertainty in projecting atmospheric river frequency has been linked to uncertainty in projecting the meridional shift of the jet position in the future (Gao et al., 2015; 2016; Hagos et al., 2016), with consequential impacts on robust predictions of regional hydrologic extremes in areas frequented by land falling atmospheric rivers.

With the high resolution simulations resolving more realistic orographic features in western North and South America and western Europe (Wehner et al, 2010), this motivates more detailed analysis of regional precipitation and hydrologic extremes including changes in the amount and phase of extreme precipitation, snowpack, soil moisture, and runoff and rain-on-snow flooding events in a warmer climate than have been attempted previously with the coarser resolution CMIP3 and CMIP5 model outputs.

## 6.5 Tropical Cyclones

Recent studies (Walsh et al., 2012; 2015; Shaevitz et al., 2014; Scoccimarro et al., 2014; Villarini et al., 2014) have highlighted the benefits of enhanced model resolution on the representation of several aspects of tropical cyclones (TCs), including the formation patterns, genesis potential index,

and the relative impact on precipitation. HighResMIP will provide an ideal framework to systematically investigate the influence of model resolution on the representation of tropical cyclones in the next generation of climate models.

It is expected that by improving the representation of the background, large-scale (oceanic and atmospheric) pre-conditioning factors affecting TC dynamics (such as wind shear and ocean stratification) via a refinement of model resolution, the overall representation of TC properties (including structure and statistics) will be affected. The potential remote influence of TCs on high-latitude processes suggested by a few authors  - e.g., TC impacts on sea-ice export in the Arctic

region (Scoccimarro et al., 2012), extra-tropical transition (Haarsma et al., 2013) and extreme precipitation events over Europe (Krichak et al., 2015) – is another (so far, poorly explored) topic that may benefit from the HighResMIP multi-model effort.

Finally, the 1950-2050 time window targeted in HighResMIP experiments will allow an evaluation of

the stationarity of the relationship between TC frequency and intensity, and the underlying, large-scale environmental conditions (Emanuel, 2015).

### 7. Additional potential applications of HighResMIP simulations

Given the relatively short time period for integration and small ensemble size, and the fact that Tier 3 simulations are also limited by using atmosphere only models, we must give careful consideration to the applications for which the HighResMIP simulations can be used.

Below is a non-exhaustive list of additional issues, not discussed in the analysis plan, that can be

addressed by HighResMIP:

1.  **Detection and attribution.** Several studies on detection and attribution of changes of weather and seasonal climate extremes would benefit from having an ensemble up to

30                2050 and for this shorter-term period the exact emission scenario chosen is not such a significant factor. Although the ensemble size of any single model will be small, it can be complemented over time, and the multi-resolution multi-model ensemble can be a starting point for assessing the occurrence of events within the distribution of the ensemble. Again, the increased resolution will likely result in more plausible and reliable

35                results.

A better assessment and attribution of the changes in extreme events that are already occurring and of near future changes will provide useful information for regional climate adaptation strategies and other users of climate model output such as infrastructure

40                investments that have a time horizon up to 30 years. The benefit relates to the increased physical plausibility and reliability of simulating the circulation-driven aspects of the weather extremes, which are more biased in coarser resolution climate models. The ensemble could aid in developing scenarios of potential future weather events to which society is vulnerable (Hazeleger et al., 2015) and used for impact studies such as

45                ecosystem studies, meteo-hydrological risks and landslides.

2.  **Time of emergence.** The same principle applies to the time of emergence studies: many studies show time of emergence (ToE) now or in the next few decades (depending on the variable and regions of course) – e.g. Hawkins and Sutton (2012). It seems

50                reasonable to assume that having high-resolution simulations could help to achieve this for large scale precipitation-related events.

3. **Decadal fluctuations.** The recent climate record contains several phases in which the global mean surface warming rate is lower in the observed record than predicted by models, and the multi-model multi-resolution ensemble might give insight in this. For instance, to reassess the possible causes for the recent global warming hiatus. In particular, the role of ocean heat uptake simulated by an eddy-permitting OGCM can be examined.

4. **Human health.** The effects of air pollution on human health is becoming a critical issue in some particular regions of complex topography. With the high horizontal resolutions and consequent detailed topographic forcing, the HighResMIP simulations may provide a useful ensemble of meteorological fields to drive either global or regional air quality modules and study the air quality effects on health.

5. **Climate services.** Climate services in different sectors such as agriculture, energy production and consumption could benefit from user-relevant diagnostics computed from high resolution future projections.

Another potential use of these simulations is to give a baseline of the forced response only (using the best estimate of the SST forced response and the SSPx radiative forcing) for near-term decadal predictions. This can then be combined with coupled decadal predictions (or statistical modelling) that also include the ocean variability and its influence. See for instance Hoerling et al. (2011) as a first attempt to do this with low-resolution models.

**8 Discussion and conclusions**

HighResMIP will for the first time coordinate high resolution simulations and process-based analysis at an international level and perform a robust assessment of the benefits of increased horizontal resolution for climate simulation. As such it is an important step in closing the gap between climate modelling and NWP, by approaching weather resolving scales. A better representation of multiple-scale interactions is essential for a trustworthy simulation of the climate, its variability and its response to time varying forcings and boundary conditions. HighResMIP thereby focuses on one of the three CMIP6 questions "what are the origins and consequences of systematic model biases?". Specifically it will investigate the relation of these model biases with small scale systems in the atmosphere and ocean and how well they are represented in climate models.

Despite the importance of enhancing horizontal resolution, many processes still have to be parameterized. For processes and regions where these parameterizations are crucial, increasing horizontal resolution did not improve the model bias. The role of various parameterizations on model biases will be investigated in other MIPs, for instance in AerChemMIP, CFMIP and RFMIP. Jointly they will address the grand challenges of the WCRP from different angles.

HighResMIP will address the grand challenges of the WCRP in the following way:

*Clouds, Circulation and Climate Sensitivity*

HighResMIP will address this grand challenge through investigating the sensitivity to increasing resolution of water vapor loading, cloud formation and circulation characteristics, with analysis concentrating on the relevant processes (see 6.3).

To improve the robustness of our understanding, the multi-model ensemble at different resolutions, together with the longer AMIP integrations, will allow us to:

|      | (i)   | link tropospheric circulation to changing patterns of SSTs, land-surface properties, and understanding the role of cloud processes in natural variability. |
|      | (ii)  | examine the extent and limits of our understanding of patterns of precipitation. |
|      | (iii) | examine changes in model biases (such as humidity) with resolution, since there |
| 5    |       | are some indications that these may be linked to climate sensitivity. |

Increasing resolution affects in particular small scale process such as the formation of clouds. Although the formation of clouds has still to be parameterized under typical resolution used within HighResMIP, the dynamical constraints for the formation of clouds, such as the location and

magnitude of upward and downward motion associated with frontal systems and orography, as well as moisture availability, are sensitive to resolution. This also applies to the response of the circulation to cloud formation.

*Changes in Water Availability*

HighResMIP is very relevant to this grand challenge. Resolution affects the hydrological cycle by modifying the land/sea partitioning of precipitation. Increasing resolution in general increases the moisture convergence over land (Demory et al., 2014) although regionally this can be reversed such as for instance in Europe during the winter due to changes in the position of the storm track (Van Haren et al., 2014). In addition, simulation of extreme precipitation events are highly sensitive to

increasing resolution. How robust are these results across the multi-model ensemble? Can higher resolution models help to give insight into inconsistencies between global precipitation and energy balance datasets? How surface water availability (P minus E) changes with warming is of significant societal relevance. HighResMIP will provide insights on uncertainty in projecting the changes as increasing model resolution alters precipitation (both amount and phase) and evapotranspiration

through changes in atmospheric circulation, land surface processes, and land-atmosphere interactions.

*Understanding and Predicting Weather and Climate Extremes*

HighResMIP is strongly related to this grand challenge. Increasing resolution of climate models will

bring us closer to the ultimate goal of seamless prediction of weather and climate. Extremes mostly occur and are driven by processes on small temporal and spatial scales that are not well resolved by standard CMIP6 climate models. Dynamical downscaling only partially resolves this limitation due to the non-linear interaction between large and small spatial scales and the importance of representing global teleconnection patterns. We aim to improve our understanding of the interaction between

global modes of variability (e.g. ENSO, NAO, PDO) and regional climate inter-decadal variability and extremes, as well as between local topographic features and the triggering of extreme events.

*Regional Climate Information*

Regional climate information focuses on smaller scales and extreme events, which are relevant for

stakeholders and adaptation strategies. This requires high resolution modeling to provide reliable information. Increasing resolution globally allows to better capture, not only local processes that could be captured by regional climate models, but also teleconnections with distant regions which could have a strong impact on the region of interest. Recent high resolution modeling studies (Di Luca et al., 2012; Bacmeister et al., 2013) and comparisons of CMIP3 and CMIP5 results (Watterson

et al., 2014) have demonstrated the added value of increased resolution for regional climate information. Model outputs from HighResMIP could also be used by the regional climate modeling community for comparison of dynamical downscaling and global high resolution approaches and for further dynamical downscaling by cloud resolving regional models and statistical downscaling for impact assessments.


*Cryosphere in a Changing Climate*

In the Tier 2 coupled simulations, the better representation of sea ice deformation, drift and leads as well as heat storage and release with increased resolution can contribute to better capturing the growth and motion of sea-ice, the air-sea heat flux, and deep-water production in polar regions, processes that are strongly affected by small scale processes. Based on HighResMIP coordinated simulations we can make a robust assessment of the effect of model resolution on Arctic sea-ice variability, including sea ice circulation and export through Fram and Davis straits, and possible influences on mid-latitude circulation. Analysis of the cryosphere in the Tier 1 experiments will, however, be somewhat limited due to the prescribed sea-ice distribution. Its main impact will be on the distribution of snow fall and subsequent accumulation and melting of the snowpack that affect land surface hydrology.

The simulations in HighResMIP will obviously be demanding with respect to High Performance Computing capability, particularly in order to complete them in a reasonable time frame. There are ongoing efforts to acquire supra-national resources in Europe and elsewhere, and also the Tianhe-2 supercomputer, one of the most powerful systems in the world, offers huge computing resources to support HighResMIP in China.

HighResMIP has evolved from the need to harmonize existing projects of high-resolution climate modelling. The European Horizon2020 project PRIMAVERA, in which major European climate centers are participating, has coordinated the initiatives for a common protocol within the CMIP6 framework. As such, the simulations conducted in PRIMAVERA will be first under the HighResMIP protocol.

It is expected that HighResMIP will be a major step forward in entering the area of weather resolving climate models and thereby opening new avenues of climate research. Fundamental new scientific knowledge is expected on weather extremes, the hydrological cycle, ocean-atmosphere interactions and multiple scale dynamics. As such, it will contribute more trustworthy climate projections and risk assessments.

**Data Availability**

The model output from the DECK and CMIP6 historical simulations will be distributed through the Earth System Grid Federation (ESGF) with digital object identifiers (DOIs) assigned. As in CMIP5, the model output will be freely accessible through data portals after registration. In order to document CMIP6's scientific impact and enable ongoing support of CMIP, users are obligated to acknowledge CMIP6, the participating modelling groups, and the ESGF centers (see details on the CMIP Panel website at http://www.wcrp-climate.org/index.php/wgcm-cmip/about-cmip). Further information about the infrastructure supporting CMIP6, the metadata describing the model output, and the terms governing its use are provided by the WGCM Infrastructure Panel (WIP) in their invited contribution to this Special Issue. Along with the data itself, the provenance of the data will be recorded, and DOI's will be assigned to collections of output so that they can be appropriately cited. This information will be made readily available so that published research results can be verified and credit can be given to the modelling groups providing the data. The WIP is coordinating and encouraging the development of the infrastructure needed to archive and deliver this information.

In order to run the experiments, datasets for natural and anthropogenic forcings are required. These forcing datasets are described in separate invited contributions to this Special Issue. The forcing datasets will be made available through the ESGF with version control and DOIs assigned.

**Table 1**: Forcings and initialization for the Historic simulations (pre-2015)

| Input | CMIP6 AMIPII | HighResMIP Tier 1 highresSST-present | Tier 2 coupled hist-1950, control-1950 |
|---|---|---|---|
| Period | 1979-2014 | 1950-2014 | 1950-2014 |
| SST, sea-ice forcing | Monthly 1˚ PCMDI dataset (merge of HadISST2 and NOAA OI-v2) | Daily ¼˚ HadISST2-based dataset (Rayner et al., 2016) | N/A |
| Anthropogenic aerosol forcing | Concentrations or emissions, as used in Historic CMIP6 simulations (Eyring et al., 2016) | Recommended: Specified aerosol optical depth and effective radius deltas from MACv2.0-SP model (Stevens et al., 2015) | Same as Tier 1 |
| Volcanic | As used in Historic | As used in Historic | Same as Tier 1 |
| Natural aerosol forcing – dust, DMS | As used in Historic | Same | Same |
| GHG concentrations | As used in Historic | Same | Same |
| Ozone forcing | CMIP6 monthly concentrations, 3D field or zonal mean, as in Historic | Same | Same |
| Solar variability | As in Historic | Same | Same |
| Imposed boundary conditions – land sea mask, orography, land surface types, soil properties, leaf area index/canopy height, river paths | Based on observations, documented. LAI to evolve consistent with land use change. | Land use fixed in time, LAI repeat (monthly or otherwise) cycle representative of the present day period around 2000 | Same as Tier 1 |
| Initial atmosphere state | Unspecified – from prior model simulation, or observations, or other reasonable ways. | ERA-20C reanalysis recommended (ideally same at high and standard resolution) | From spin-up of coupled model in 3.2.1 |
| Initial land surface state | Unspecified – as above. May require several years of spin-up, cycling 1979 or starting in early 1970s | ERA-20C reanalysis recommended, spun-up in some way | From spin-up |
| Ensemble number | Typically >=3 | >= 1 | 1 |
| Initial ocean/sea-ice state | N/A | N/A | From coupled spin-up |

**Table 2**: Forcings for the future climate simulations

| Input | High end CMIP6 SSPx Scenario (ScenarioMIP) | HighResMIP Tier 3 highresSST-future | Tier 2 coupled highres-future |
|---|---|---|---|
| Period | 2015-2100 | 2015-2050 | 2015-2050 |
| SST, sea-ice forcing | N/A | Blend of variability from ¼° HadISST2-based dataset (Rayner et al., 2016) and climate change signal from CMIP5 RCP8.5 models | N/A |
| Anthropogenic aerosol forcing | Concentrations or emissions (ScenarioMIP) | Specified aerosol optical depth and effective radius deltas from MACv2.0-SP model | Same as Tier 3 |
| Natural aerosol forcing – dust, DMS | ScenarioMIP | Same as Tier 1 | Same |
| Volcanic aerosol | ScenarioMIP | Volcanic climatology | Same as Tier 3 |
| GHG concentrations | ScenarioMIP SSPx | SSPx | Same as Tier 3 |
| Ozone forcing | CMIP6 monthly concentrations, 3D field or zonal mean, 2015-2100, based on SSPx ScenarioMIP | Same | Same |
| Solar variability | CMIP6 dataset | Same | Same |
| Imposed boundary conditions – land sea mask, orography, land surface types, soil properties, leaf area index/canopy height, river paths | Based on observations, documented.  LAI to evolve consistent with land use change. | Land use fixed in time, LAI repeat (monthly or otherwise) cycle | Same as Tier 1 |
| Initial atmosphere, ocean, sea-ice state | Continuation from Historic simulation | Continuation from Tier 1 simulation | Continuation from Tier 2 historic simulation |
| Ensemble number | Typically >=3 | >= 1 | 1 |

## 9 Appendix
### 9.1 Participating models in HighResMIP

**Table 9.1**: Model details from groups expressing intention to participate in at least Tier 1 simulations, together with the potential model resolutions (if known/available, blank if not).

| Model name | Contact Institute | Atmosphere Resolution (STD/HI) mid-latitude (km) | Ocean Resolution (HI) |
|---|---|---|---|
| AWI-CM | Alfred Wegener Institute | T127 (~100 km) T255 (~50 km) | 1-1/4 degree 0.05-1 degree |
| BCC-CSM2-HR | Beijing Climate Center | T106 (~110 km) T266 (~45 km) | 1/3-1 degree |
| BESM | INPE | T126 (~100 Km) T233 (~60 Km) | 0.25 degree |
| CAM5 | Lawrence Berkeley National Laboratory | 100 km 25 km | |
| CAM6 | NCAR | 100 km 28 km | |
| CMCC | Centro Euro-Mediterraneo sui Cambiamenti Climatici | 100 km 25km | 0.25 degree |
| CNRM-CM6 | CERFACS | T127(~100km) T359(~35km) | 1 degree 0.25 degree |
| EC-Earth | SMHI, KNMI, BSC, CNR and 23 other institutes | T255(~80km) T511/T799(~40/25km) | 1 degree 0.25 degree |
| FGOALS | LASG, IAP, CAS | 100 km 25 km | 0.1-0.25 degree |
| GFDL | GFDL | 200 km - | |
| INMCM-5H | Institute of Numerical Mathematics | - 0.3 x 0.4 degree | 0.25 x 0.5 degree 1/6 x 1/8 degree |
| IPSL-CM6 | IPSL | 0.25 degree | |
| MPAS-CAM | Pacific Northwest National Laboratory | - 30-50km | 0.25 degree |
| MIROC6-CGCM | AORI, Univ. of Tokyo/JAMSTEC/National Institute for Environmental Studies (NIES) | - T213 | 0.25 degree |
| NICAM | JAMSTEC/AORI/ The Univ. of Tokyo/RIKEN/AICS | 56-28 km 14km (short term) | |
| MPI-ESM | Max Planck Institute for Meteorology | T127(~100km) T255(~50km) | 0.4 degree |
| MRI-AGCM3 | Meteorological Research Institute | TL159(~120km) TL959 (~20km) | |
| NorESM | Norwegian Climate Service Centre | 2 degree 0.25 degree | 0.25 degree |
| HadGEM3-GC3 | Met Office Hadley Centre | 60km 25km | 0.25 degree |

**9.2 Future SST and sea-ice forcing**

Discussion with the HighResMIP participants suggests that the agreed approach is to use the RCP8.5 scenario, and use the CMIP5 models to generate the projected future trend. Numerical code for the following calculations will be made available in Python, as will the final dataset on the ¼ degree daily HadISST2.2.0 grid.

So following Mizuta et al (2008) for the most part, the algorithm is as follows:
For HadISST2.2.0 (Rayner et al., 2016) in the period 1950-2014:

For each year y, month m, and grid point j:
Calculate the time mean, monthly mean $T_{mean}(m, j)$
Calculate the linear monthly trend $T_{trend}(m, j)$ over the period
And then the interannual variability $T_{var}$ as the residual:
$T_{HadISST2}(y, m, j) = T_{mean}(m, j) + T_{trend}(m, j) + T_{var}(y, m, j)$

Then from at least 12 CMIP5 coupled models during the period 1950-2100 (using the Historic and RCP8.5 simulations).
Calculate a monthly mean trend, for each model over this period, as a difference from several years centred at 2014, so that the change in temperature can be smoothly applied to the HadISST2 dataset.
$T_{model\_trend}(y, m, j) = T_{model}(y,m,j) - T_{model}(mean(2004-2024), m, j)$.

Regrid this trend to the HadISST2 1/4 degree grid.
Calculate the multi-model ensemble mean of this monthly trend.
$T_{multi\_trend}(y,m,k) = ensemble\ mean(T_{model\_trend})$

This ensemble mean still contains a large component of both spatial and temporal variability – since the object here is to produce a large-scale, smoothly varying background signal to the HadISST2 variability, this multi-model trend is spatially filtered (using a 20x10 longitude-latitude degree box car filter), and temporally filtered using a Lanczos filter with a 7 year timescale.

Then for the future period, the temperature is:
$T_{future}(y, m, j) = T_{mean}(m, j) + T_{var}(y, m, j) + T_{multi-trend}(y,m,k)$

This will repeat the variability from the past period into the future, but adding the model future trend. The choice of 1950 as a start date for this section is that it has the most similar phase of some of the major modes of variability (AMO, PDO etc) to use for the repeat.

```
HadISST2: 1870-----------------------------1950----------- 2014
Cut out a section                          |-----------------------|
Concatenate this section (twice) to the end of HadISST2 at 2014:
HighResMIP_ISST: 1850-------------1950----------2014|----------------|2078|----------|2100
```

Projecting the sea-ice into the future will be based on the following procedure:
1. Using observed SST and sea ice concentration an empirical relationship is constructed (HadISST2 (Rayner et al., 2016) uses the inverse method to derive SST based on sea-ice concentration).

This is done by dividing the SST into bins of 0.1K. The SST of each data point determines in which bin the sea ice concentration of each data point falls. After all data points are handled in this way the mean sea-ice concentration for each bin is computed. The relationship is different for the Arctic and Antarctic and seasonally dependent.

2. Using this empirical relationship between SST and Sea-ice concentration the sea-ice concentrations for the constructed SST are computed.

However, a couple of alternative methods are also being investigated, such as that used in HadISST2 (Titchner and Rayner, 2014), in which the sea-ice edge is located, and then the concentration is filled

in from here towards the pole.

### 9.3 Targeted additional experiments
#### 9.3.1 Leaf Area Index (LAI) experiment – *highresSST-LAI*

The LAI is one of the most common vegetation indices that describe vegetation activity (Chen and Black, 1992). It closely modulates the energy balance, as well as the hydrological and carbon cycles of the coupled land-atmosphere system at different spatiotemporal scales (Mahowald et al., 2015). For atmosphere-ocean GCMs, including those of HighResMIP, the mean seasonal cycle of LAI is commonly prescribed to improve the physical and biophysical simulations of the land-atmosphere

system (Taylor et al., 2011). To reduce the potential uncertainties due to inconsistent LAI inputs for different models participating in HighResMIP, we propose to conduct targeted LAI experiments, with a common LAI data set.

Various remote sensing based LAI datasets have been recently developed (Fang et al., 2013; Zhu et

al., 2013). Among them, the LAI3g data has been found to be the best, in terms of continuity, quality and extensive applications (Zhu et al., 2013; Mao et al., 2013). For the targeted experiments we will provide a ¼ degree mean LAI3g data set. The other boundary conditions (e.g., greenhouse gases and aerosols, SST and Sea-Ice conditions) will be identical to those in the Tier 1. The new targeted simulations will be directly compared to the Tier 1 results, for which each modeling center has used

their preferred LAI.  If significant positive impacts are found, then the next CMIP might consider applying LAI3g as a new common high-resolution LAI dataset.

#### 9.3.2 Impact of SST variability on large scale atmospheric circulation – *highresSST-smoothed*
The impact of mesoscale air-sea coupling on the large-scale circulation (in atmosphere and ocean) is

a growing area of research interest. Ma et al. (2015) have shown that mesoscale SST variability in the Kuroshio region can exert an influence on rainfall variability along the U.S. Northern Pacific coast. In order to assess this, we propose parallel simulations of the high resolution ForcedAtmos model using spatially filtered SST forcing.

The modeling approach is to conduct twin-experiments - one with high resolution SST (the reference HighResMIP simulation) and another with spatially low-pass filtered SST. This approach appears to be quite effective in dissecting the effect of mesoscale air-sea coupling. The filter should be the LOESS filter used by Ma et al (2015) and Chelton and Xie (2010). The parallel simulation should start in 1990 from the HighResMIP simulation and be identical apart from the SST forcing.

Period of integration: 10 years. This should be done in an ensemble multi-model approach to ensure statistically significant results.

#### 9.3.3 Idealized forcing experiments with CFMIP – *highresSST-p4K, highresSST-4co2*
CFMIP experiments using +4K and 4xCO2 perturbations are used to evaluate feedbacks, effective

radiative forcing and rapid tropospheric adjustments (e.g. to cloud and precipitation).  Although the horizontal resolutions used by most groups within HighResMIP do not approach the cloud-system

resolving scale (and hence may not be expected to generate a significantly different response), there is potential for differences in response at the regional scale.

Period of integration: 10 years for each +4K and 4xCO2 (in parallel to the 2005-2014 HighResMIP simulation period for best comparison with recent observations).

### 9.3.4 Abrupt forcing in coupled experiments with CFMIP and OMIP – *highres-4co2*

CFMIP experiments use abrupt 4xCO2 forcing in a piControl experiment to look at ocean heat uptake. We will similarly do abrupt 4xCO2 at the end of the spinup period of the control-1950 simulations for each coupled model resolution, to study the impact of the ocean resolution on heat uptake. This experiment has the added benefit of further investigation of spin-up processes.

Period of integration: 20 years (in parallel with the first 20 years of control-1950, after the initial spin-up period).

### 9.3.5 Tier 2 and 3 using RCP8.5 instead of SSPx - *highres-RCP85*

This option is included for centers, such as those involved in the European H2020 project PRIMAVERA, that have to start their simulations before the availability of SSPx. It is motivated by the notion that the differences between SSPx and RCP8.5 will be limited up to 2050. If in a joint analysis the SSPx and RCP8.5 ensembles appear to be significantly different, than the RCP8.5 centers are recommended to repeat their simulations with SSPx, which, due to the short integration period of 36 years, should not be prohibitive.

### Acknowledgements

PRIMAVERA project members (M. Roberts, R. J. Haarsma, P. L. Vidale, T. Koenigk, V. Guemas, S. Corti, J. Von Hardenberg, J-S von Storch, W. Hazeleger, C.A. Senior, M. Mizielinsky, T. Semmler, A. Bellucci, E. Scoccimarro, N. S. Fučkar ) acknowledge funding received from the European Commission under Grant Agreement 641727 of the Horizon 2020 research programme.

C. Kodama acknowledges Y. Yamada, M. Nakano, T. Nasuno, T. Miyakawa, and H. Miura for analysis ideas.

N.S. Fučkar acknowledges support of the Juan de la Cierva-incorporación postdoctoral fellowship from the Ministry of Economy and Competitiveness of Spain.

L. R. Leung and J. Lu acknowledge support from the U.S. Department of Energy Office of Science Biological and Environmental Research as part of the Regional and Global Climate Modeling Program. The Pacific Northwest National Laboratory is operated for DOE by Battelle Memorial Institute under Contract DE-AC05-76RLO1830.

J. Mao is supported by the Biogeochemistry-Climate Feedbacks Scientific Focus Area project funded through the Regional and Global Climate Modeling Program in Climate and Environmental Sciences Division (CESD) of the Biological and Environmental Research (BER) Program in the U.S. Department of Energy Office of Science. Oak Ridge National Laboratory is managed by UT-BATTELLE for DOE under contract DE-AC05-00OR22725.

We thank Martin Juckes and his team for all their work on the HighResMIP and CMIP6 data request. Nick Rayner and John Kennedy for allowing early access to the HadISST2 daily, 1/4 degree SST and sea-ice dataset. Mark Ringer and Mark Webb for ideas for the targeted CFMIP-style experiment. Francois Massonnet for discussions on high resolution modelling and sea-ice.

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
