# Peer review of "High Resolution Model Intercomparison Project (HighResMIP v1.0) for CMIP6"

_Geoscientific Model Development, 2016_

## Short Comment (SC1) · 13 Apr 2016

Dear authors,

In agreement with the CMIP6 panel members, the Executive editors of GMD would like to establish a common naming convention for the titles of the CMIP6 experiment description papers.

The title of CMIP6 papers should include both the acronym of the MIP, and CMIP6, so that it is clear this is a CMIP6-Endorsed MIP.

Additionally, we strongly recommend to add a version number to the MIP description. The reason for the version numbers is so that the MIP protocol can be updated later, normally in a second short paper outlining the changes. See, for example:

[Figure]

http://www.geosci-model-dev.net/special_issue11.html,

Good formats for the title include:

'XYZMIP (v1.0) contribution to CMIP6: Name of project'

or

'Name of Project (XYZMIP v1.0) contribution to CMIP6'

If you want to include a more descriptive title, the format could be along the lines of,

'XYZMIP (v1.0) contribution to CMIP6: Name of project - descriptive title'

or

'Name of Project (XYZMIP v1.0) contribution to CMIP6: descriptive title.'

When you revise your manuscript, please correct the title of your manuscript accordingly.

Yours,

Astrid Kerkweg
* * *

---

## Short Comment (SC2) · 18 Apr 2016

Table 1 and 2 of the paper suggest that the volcanic forcing is from MACv2-SP. However, MACv2-SP is only providing anthropogenic aerosols. The volcanic forcing in High-ResMIP should be handled in a similar way as in the other MIPs: use prescribed stratospheric aerosols, see http://www.wcrp-climate.org/wgcm-cmip/wgcm-cmip6 for details.

---

## Short Comment (SC3) · 19 Apr 2016

There is an inconsistency in the suggested GHG forcing (concentrations) between the historical and future run. For the historical run it is suggested to use the new data for CMIP6 but for the future run it will be the old RCP8.5 scenario from CMIP5 (Table 1 and 2). This inconsistency causes two problems:

1) It is more difficult to have a model version that can handle both the old and new foricng datasets at the same time. It would be pretty easy if it would be just a textfile that changes when going from CMIP5 to CMIP6 forcing, but thsi is not the case. For CMIP6 it is recommended that the models use monthly varying zonal distributions or at least separate GHG concentrations in the two hemispheres (see http://www.climate-energy-college.net/search/content/cmip6). For CMIP5 on the other hand only global

annual means have been prescribed. Implementing both options for 1-d and 3-d forcing is not impossible yet certainly nothing positive.

2) How consistent are the CMIP5 and CMIP6 forcings? I can imagine that there will be a change in GHG concentrations if the new CMIP6 data are used until 2014, and the old data afterwards. We cannot tell because the future GHG emissions aren't available at this stage, but I am pretty sure that the creators of the dataset will check that historical and future scenarios of the CMIP6 datasets match, but I doubt that they check if historical from CMIP6 and future from CMIP5 datsets fit well together.

I would suggest that the GHG forcing is kept consistent across the historical and future simulations (as it is to be done in any other MIP). Either we take the CMIP6 GHG forcing for both phases (preferentially), or the old RCP8.5 from CMIP5, but not a blend of the two.

---

## Short Comment (SC4) · 6 May 2016

In order to be consistent with the naming conventions for CMIP6, need to change: highres-LAI to highresSST-LAI

highres-p4k to highresSST-p4k

highres-4co2 to highresSST-4co2

We recommend that the initial condition for the atmosphere and land for 1950 (for the highresSST-present, and perhaps the highres-1950 experiment) come from the ERA-20C reanalysis from January 1950.

---

## Referee Comment (RC1) · Anonymous Referee #1 · 20 May 2016

This manuscript describes the proposed HiResMIP protocol which focused on the inter-comparison of "high" resolution AGCMs and CGCMs, defined as 25 to 50 km resolution for the atmosphere and eddy permitting for the ocean.

Main comments:

1. My main comment is that the current draft does not make a compelling case as why such a comparison is needed and what are the expected scientific benefits. Many are claimed but the current text does not justify them in a clear way.

2. For example a major claim is that such a MIP will help model improvements. But I could not find in practice what will this mean, i.e. how the knowledge obtain will inform model development. Increasing the resolution has always been a natural pathway for modelling groups and it is unclear how having this MIP or not will change the related

priorities.

3. Another example is that only horizontal resolution is included in the protocol. This is a very serious shortcoming as many processes depend on both resolutions (horizontal and vertical) such as atmospheric convection or ocean mixing. As quickly alluded to, solving the diurnal cycle over the ocean requires 1m vertical resolution at the top level of the ocean and 2-3 hours coupling time step. I was surprised that this is not a requirement for this MIP.

4. The introduction suggests that modes of interannual variability can be analysed in these short simulations, which is not the case for ENSO for example (several centuries are needed). The details given near the end are careful not to include the modes prone to this sampling issue but the introduction should clarify these limits upfront.

5. The forcings are going to be different between the CTRL and the HiRes simulations for some model (e.g. aerosols). This is an issue that will prevent a clean comparison. Along the same lines, when changing the resolution one can never have "exactly the same parameters". This limit also renders the comparison less informative.

6. The use of daily SSTs for the AMIP simulation is an issue I believe. As shown by several studies (Wu and Kirtman 2005, 2007, Cassou 2015), the mid-latitudes ocean is forced by the atmosphere, not the other way around. The classical use of smoothed monthly SSTs somewhat mitigates this problem. The use of daily SST requires a much better justification and an analysis that this will not have a impact on extremes over land (adverse impact was shown by Cassou 2015).

7. Finally the discussion on the benefits of increased resolution is not balanced, and mostly ignores the studies that don't show any impact of resolution, for example on model biases. The role of physical parameterization is not discussed even though it is central. Improving this balance would strengthen the manuscript which currently mostly appears as a manifesto of like-minded people.

References:

Cassou C. (2015). Some critical technical choices for pacemaker experiments, Aspen DCPP Workshop June 2015. (http://www.docfoc.com/aspen-dcpp-workshop-june-2015-some-critical-technical-choices-for-pacemaker)

Wu, R., Ben P Kirtman, Pegion, K., Center for Ocean-Land-Atmosphere Studies. (2005). Local air-sea relationship in observations and model simulations.

Wu, R., & Kirtman, B. P. (2007). Regimes of seasonal air-sea interaction and implications for performance of forced simulations. Climate Dynamics, 29(4), 393–410. http://doi.org/10.1

————————————————

---

## Referee Comment (RC2) · Anonymous Referee #2 · 31 May 2016

General comments ——————

This paper presents the rationale and experimental design of the internationally coordinated experiments of the intercomparison project HighResMIP proposed within the framework of the WCRP CMIP6 experiments. This set of experiments aim at investigating the role of spatial resolution with the objective to evaluate the impact of very high resolution (from 50 down to 25 kms). The paper presents very well the state of the art and how much previous work allows expecting improvements in the representation of small to large-scale phenomena. The ensemble of simulations from different climate models is expected to provide a robust estimate of the impact of high resolution. The experimental design is divided in three tiers including atmospheric alone models and coupled atmosphere-ocean models and complemented by some possible additional sensitivity experiments. Within the CMIP6 framework, HiResMIP will bring insights into

the specific issue of better understanding the origin and consequences of systematic model biases as well as the grand challenges associated with clouds, water availability and climate extremes. The paper is needed to describe the experimental design of HighResMIP. In overall it is clear and promising, although some aspects need to be improved and some redundancies reduced (see specific comments).

Specific comments ———————

Introduction: in the first paragraph describing the state of the art on the impact of resolution, the authors should be more explicit on which resolution are concerned by the cited experiments. In the second paragraph, it would be good to know how many coupled models have been run at very high resolution.

Section 2: The 3 Tiers are clearly defined but it is not clear whether there is a ranking behind: are Tier 1 experiments a pre-requisite minimum to participate to HiResMIP ?

How HighResMIP is linked to CMIP6 (sections 2., 4.1 and 4.2) is ambiguous: since the very high resolution are not run under the DECK conditions (4.1), I guess only the standard resolution is supposed to be in CMIP6 and the very high resolution are complementary to CMIP6 but not really part of CMIP6 (in 2. It is said "linked" but in section 4.2 it says HiResMIP as one of the endorsed MIPs)? Please clarify.

Section 2, Page 5, second paragraph on Tier 2: coupled runs are only mentioned as an opportunity to understand the role of natural variability but they are also required to investigate future climate change.

Section 2, Page 5, lines 21 to 26: the lack of tuning will most probably be more critical for coupled runs than for atmospheric alone models which are constrained by a fixed SST. This should be mentioned.

Section 2, Pages 5 and 6: For clarity I would recommend to put the sections on common forcing fields (2.1, 2.1.1, 2.1.2, 2.1.3) together with the description 3.1 of the Tier 1. For other Tiers, reference to Tier 1 is then sufficient.

[Figure]

There are some redundancies between section 4.2 on links with other MIPs, section 6 on applications and section 7 on analysis plan. I understand the need for some redundancies but I have the feeling the order of arguments could be optimised. For example, it would be clearer to first describe the analysis plan, emphasizing at the same time the related MIPs (eg CORDEX, CFMIP, GMMIP) and some applications related to analyses, and then the list of other potential applications. For the interactions with other MIPS, they could be spread among the analyses and the description of the additional experiments. A table could summarise all these interactions.

Section 5, page 11, on data: In this part, it is not clear whether the plans for the list of variables to be stored is already fixed or not. Please clarify. It would also be good if an order of magnitude of the storage needed could be given. What is meant by the "design of CORDEX will be taken into account"?. What are priorities 1, 2 and 3 ?

Section 6: lines 24-25: Tier 3 experiments are also limited by using atmosphere only models.

Page 18, the data availability part should not be in this section but rather with section 5.

Technical comments ———————-

Page 4, line 11: results rather than representation ?

Page 4, lines 30-32: should be more explicit on which resolution.

Page 5, lines 10-11: the list of model names do not correspond to the list of models used in the references: models MIROC, GFDL, SINTEX-F2 whereas in the references SINTEX-F2, GFDL, Hadley, CESM

Page 5, line 34, please mention explicitly RCP8.5.

Section 2, Page 5, lines 41-42: the use of a delta to the climatological forcing is not clear enough.

Page 6, Figure 1: would be good to add 1, OA and A for Tiers 1, 2 and 3 respectively on the graphs.

Page 7, line 20. It would be good to add here that 3 runs are recommended but not mandatory (rather than page 8 lines 40-43)

Page 7, line 36-37: unclear. Is it for detection/attribution ?

Page 8, line 6: what is EN4 ?

Page 8, lines 34-38: has already been said

Section 3.2.1, Page 9, 3.2.4 is the use of EN4 recommended ? not fully clear

Page 12, line 12 the European Copernicus Climate Data Store

Page 13, line 22: rather the air quality than the aerosol only effect on health ?

Page 13, line 32: very few models will use eddy-permitting ocean. This is misleading.

Table 1: give a reference to "Historic" boundary conditons. Tier 2 future add coupled.

Table 9.1: what is the standard resolution of NorESM ? missing information for GFDL.

---

## Referee Comment (RC3) · Anonymous Referee #3 · 6 Jun 2016

The manuscript is a rather matter of fact description of the CMIP6 endorsed high-resolution MIP. As such it will form an important reference for those undertaking CMIP6 studies. With this in mind I feel positively inclined toward recommending acceptance.

My only concern is the lack of a coupled experiment that is likely to have a large signal to noise ratio. An example might be a quadrupling of $CO_2$ at some point during the 1950s control. A similar experiment is proposed in CMIP6 (albeit from pre-industrial conditions). The role of high ocean resolution in heat uptake could provide some results that may be difficult to tease out of the tier 2 simulations. The computation costs could be relatively modest. For instance, Kuhlbrodt et al. (2015) showed useful results from a 20-year experiment.

Minor comments

Abstract: define the acronym MIP

Introduction: A few more acronyms to define, e.g. ITCZ, MJO, QBO.

Section 2.1: Give a proper reference to the CMIP6 publication in this same special issue.

Figure 1: Provide a more detailed caption.

Section 3.1.1: As noted we do not have high resolution data for the entire historical period. Are there any issues with blending the pre and post satellite era data? Whatever methods are used to produce 1/4 degree SSTs, the raw observations simply are not there. For instance the process outlined for producing future (2015-2050) SSTs relies upon the variability being unchanged in a changing climate. Please comment.

Section 4.2, page 10, line 34: insert a space somewhere in andreanalysis.

Section 6, page 12, last line: No need to define ToE as acronym not used.

Section 6, Ocean model biases: insert the word "coastal" before upwelling as the equatorial upwelling zone bias is often different to this.

Section 7.1, page 14, 2nd to last sentence: Is there any evidence that sea ice simulations might improve with increasing resolution. If so give a reference. The project will be moving towards the limit of where the continuum hypothesis is reasonable, which may be an issue.

Section 7.1, page 14, last sentence: This doesn't sound quite right. Maybe Differences would be better than Difference.

Section 7.1, last sentence: replace "such as" with "outlined by" or something similar.

Section 7.4, page 16, paragraph 2: Too many acronyms reduce the readability of manuscripts. AR seems a bit unnecessary as it replaces just two words and sonly used twice. TC (a bit later on) also seems a bit unnecessary, but as it is used a few

more times I could live with it.

End of Review

---

## Author Comment (AC1) · 4 Jul 2016

Response SC1 by Astrid Kerkweg

According to your suggestions we have changed the title of the manuscript in:

High Resolution Model Intercomparison Project (HighResMIP v1.0) for CMIP6

---

## Author Comment (AC2) · 4 Jul 2016

Response to SC2 by Klaus Wyser:

"Table 1 and 2 of the paper suggest that the volcanic forcing is from MACv2-SP. However, MACv2-SP is only providing anthropogenic aerosols. The volcanic forcing in High-ResMIP should be handled in a similar way as in the other MIPs: use prescribed strato-spheric aerosols, see http://www.wcrp-climate.org/wgcm-cmip/wgcm-cmip6 for details."

Thank you for pointing out this error. We have corrected this in the revised manuscript.

---

## Author Comment (AC3) · 4 Jul 2016

Response to anonymous referee #1

"Main comments: 1. My main comment is that the current draft does not make a compelling case as why such a comparison is needed and what are the expected scientific benefits. Many are claimed but the current text does not justify them in a clear way."

We simply do not agree with this assertion. There are a large number of references in the manuscript which describe the impact of model resolution in single model or small-scale comparisons, and which also try to ascribe such changes to simulated processes and their representation. This can be extremely difficult using just one model, since it is not possible to discover if it may be caused by particular aspects of that model. The

described protocol will, for the first time, enable a systematic evaluation of the impact of model horizontal resolution. The examples of CMIP3 and CMIP5 have demonstrated the tremendous advantages of a common protocol that enables a clear comparison between the models and improved understanding of the underlying physical processes.

"2. For example a major claim is that such a MIP will help model improvements. But I could not find in practice what will this mean, i.e. how the knowledge obtain will inform model development. Increasing the resolution has always been a natural pathway for modelling groups and it is unclear how having this MIP or not will change the related priorities."

We are slightly unsure what you mean here. The manuscript does not claim to help model improvements (please indicate specifically where if you think so) – it simply sets out a common protocol to provide the framework for understanding model differences due to horizontal resolution, and to ascribe these to process representation where possible. It will help to clarify which processes will benefit from increased horizontal resolution and how this will affect the model's climate, natural variability and the response to global warming. This will help in a better understanding of these processes that will ultimately result in improved representation, including in models of lower resolution.

We agree that increasing resolution is a natural pathway for modelling groups. However, an examination of the change in horizontal resolution of many models over the last few CMIP exercises reveals that it is much less emphasized compared to complexity – our aim is to examine whether that is the correct choice or not. The advantage of a common protocol is that the impact for an individual modelling center of their simulations with increased resolution will strongly increase, because their simulation will be analyzed by many researchers and compared with other high resolution simulations. In addition the standardized output following the HighResMIP/CMIP6 protocol will strongly facilitate the analysis. The fact that already 17 centers have expressed their intention to participate in the HighResMIP simulations indicates the strong appeal of a common protocol for high resolution simulations.
"3. Another example is that only horizontal resolution is included in the protocol. This is a very serious shortcoming as many processes depend on both resolutions (horizontal and vertical) such as atmospheric convection or ocean mixing. As quickly alluded to, solving the diurnal cycle over the ocean requires 1m vertical resolution at the top level of the ocean and 2-3 hours coupling time step. I was surprised that this is not a requirement for this MIP."

We agree that the correct representation of physical processes depend on both resolutions. The scaling between horizontal and vertical resolution must obey N/f, where N is the Brunt-Väisälä frequency and f the coriolis parameter. This implies a factor of 100, between horizontal and vertical resolution. This is well satisfied by the model configurations in the HighResMIP group. We therefore focused on the increase in horizontal resolution, which we consider as the most limiting factor for many processes and makes this comparison tractable. From a practical standpoint, changing vertical resolution can be extremely complex (due to many interactions with parameterisations particularly in the atmosphere) and in our opinion more likely to be dependent on individual model choices, and hence does not lend itself to a clean intercomparison (at least at this time). In addition the combination of increased horizontal and vertical resolution would complicate to assign the contribution of each of those. We have added these arguments in the text (line 4-11 page 4.).

"4. The introduction suggests that modes of interannual variability can be analysed in these short simulations, which is not the case for ENSO for example (several centuries are needed). The details given near the end are careful not to include the modes prone to this sampling issue but the introduction should clarify these limits upfront."

This is a good point. We do not want to overstate this and we agree that this should be clarified already in the introduction. This is now discussed in lines 32-35 page 3.

"5. The forcings are going to be different between the CTRL and the HiRes simulations for some model (e.g. aerosols). This is an issue that will prevent a clean comparison.

Interactive
comment

Along the same lines, when changing the resolution one can never have "exactly the same parameters". This limit also renders the comparison less informative."

We agree that with built-in scale dependence of some parameters, one never can have "exactly the same parameters". We also agree that due to interpolation the forcings at different resolutions will be somewhat different (but as close as is currently possible). However, simply because the comparison is not perfect does not lessen the amount we can learn. As long as we can account for these inherent difficulties, we have the opportunity to find out if the impact of enhanced resolution is robust across models, given a common protocol. This has been clearly shown by the analyzes of the already existing high resolution simulations, discussed in the introduction. In addition these obstacles for a clean comparison will likely deviate between the models.

"6. The use of daily SSTs for the AMIP simulation is an issue I believe. As shown by several studies (Wu and Kirtman 2005, 2007, Cassou 2015), the mid-latitudes ocean is forced by the atmosphere, not the other way around. The classical use of smoothed monthly SSTs somewhat mitigates this problem. The use of daily SST requires a much better justification and an analysis that this will not have a impact on extremes over land (adverse impact was shown by Cassou 2015)."

The temporal resolution of SST in AMIP runs is indeed an issue. In AMIP runs the ocean has an infinite heat capacity. This has a deleterious impact on the phase relationships between SSTs, overlying atmosphere, and surface fluxes (Barsugli and Battisti, 1998). This occurs also on monthly time scales as outlined by Sutton and Mathieu (2002). Indeed there is in the mid-latitudes a strong forcing of the ocean by the atmosphere, however, recent studies (Minobe et al., 2008; Kirtman et al, 2012; Parfitt et al., 2015; Ma et al., 2015; O'Reilly et al., 2015) revealed that there is also a significant forcing of atmosphere by the ocean especially along ocean fronts, with sharp temperature gradients and energetic mesoscale eddy activities that are collocated in the genesis regions of the storm tracks. A correct simulation of these processes requires that the strong SST gradients and mesoscale eddies are resolved. This implies the use of daily
data, because due to the strong meandering of the western boundary currents, time averaging will strongly smooth the SST fields. Because we focus in HighResMIP on the impact of horizontal resolution and how this affects the small scale processes we therefore will use daily, 0.25 degree SSTs.

In conclusion we state that due to the fundamental problems with AMIP runs, there is no general preferred time scale for averaging SSTs although for certain aspects and processes of the climate system the problem will somewhat mitigated by time averaging as explained by you. On the other hand time averaging will mask important processes that we hypothesize to be resolution dependent and therefore a focus of HighResMIP. We have added a discussion on the use of daily SSTs on page 8, lines 7-12.

We will make use of the DECK AMIP simulations, as well as our smoothed SST experiment, to better understand the impact of higher resolution and frequency SSTs. Most modeling groups typically use similar SST datasets (OI-SST, ESA CCI, ERA-Interim) for research purposes, particular as model resolution is enhanced, and hence one expected outcome of HighResMIP is an indication of the strengths and weaknesses of such an approach.

"7. Finally the discussion on the benefits of increased resolution is not balanced, and mostly ignores the studies that don't show any impact of resolution, for example on model biases. The role of physical parameterization is not discussed even though it is central. Improving this balance would strengthen the manuscript which currently mostly appears as a manifesto of like-minded people."

We realize that this manuscript is indeed written by researchers that support the idea of the added value of high resolution runs. This is part of how the new CMIP phase is organized along different specialized MIPs to address the great challenges of the WCRP. We subscribe the hypothesis that high resolution simulations will contribute in resolving those challenges. We are fully aware that other researchers may be more skeptical. Only by doing these experiments will we learn if our hypothesis is justified.

At this moment there are sufficient centers and researchers that will participate in these experiments of HighResMIP to produce a much more robust understanding of resolution impacts across a multi-model ensemble, and from this evidence and model output the community will be able to draw its own conclusions.

Although we completely agree with the reviewer about the important role of physical parameterization we will not focus on this in the manuscript. The purpose of this GMD paper is to motivate and outline the high resolution simulations, not to give an overview of the causes of model errors. Other GMD papers in this issue will focus on the role of physical parameterizations (e.g. AerChemMIP, C4MIP and RFMIP) and completely ignore the role of horizontal resolution. We think that is fair and combined this issue will provide an overarching view of the different approaches within the scientific community to address the great challenges of the WCRP. To make this clear we have added in the discussion a section about the importance of parameterization and that it will be the central topic in other MIPs on page 17, line 37-41.

---

## Author Comment (AC4) · 4 Jul 2016

Response to anonymous referee #2.

We thank the reviewer for the positive evaluation of HighResMIP and the GMD paper describing the rationale and the protocol. Below we will answer in detail the specific comments.

"Section 2: The 3 Tiers are clearly defined but it is not clear whether there is a ranking behind: are Tier 1 experiments a pre-requisite minimum to participate to HiResMIP ?"

This was indeed not specified in the protocol. It was assumed that all centers participating in HighResMIP would at least do the Tier 1 experiments, because these are considered the least demanding. We have made this now explicit, by requiring that

the Tier 1 experiments are a pre-requisite to participate to HighResMIP (page 5, line 15). The Tier 1 experiments are considered the core of HighResMIP that would benefit from an as large as possible ensemble. In addition the value of the Tier 2 coupled experiments would be strongly enhanced if they can could be compared directly with the Tier 1 uncoupled experiments.

"How HighResMIP is linked to CMIP6 (sections 2., 4.1 and 4.2) is ambiguous: since the very high resolution are not run under the DECK conditions (4.1), I guess only the standard resolution is supposed to be in CMIP6 and the very high resolution are complementary to CMIP6 but not really part of CMIP6 (in 2. It is said "linked" but in section 4.2 it says HiResMIP as one of the endorsed MIPs)? Please clarify."

Indeed, due to computational limitations, the DECK simulations will not be performed by the high resolution version. The standard resolution version, will however complete the DECK simulations and be part of CMIP6. The high resolution version is considered as a sensitivity experiment with respect to horizontal resolution. This is the overarching scientific question of HighResMIP.

"What is impact of enhanced horizontal resolution on the model characteristics?".

The standard resolution DECK simulations are considered as the entry card for High-ResMIP. Because NWP centers often only can perform SST prescribed simulations, they cannot complete the DECK simulations. We still want to offer them the possibility to participate in HighResMIP. The solution as outlined in the manuscript is that they can still participate in HighResMIP but their simulations will only be visible as HighResMIP and not as CMIP6 runs. We have tried to clarify this more in the revised manuscript and hopefully make it less ambiguous by deleting the expression "linked" in sections 2, 4.1 and 4.2

"Section 2, Page 5, second paragraph on Tier 2: coupled runs are only mentioned as an opportunity to understand the role of natural variability but they are also required to investigate future climate change."

Indeed, although a large part of the focus of HighResMIP is on natural variability, the extension of simulations up to 2050 is motivated by exploring the impact of high resolution on future climate. We have added that in the manuscript (page 5, line 18-19).

"Section 2, Page 5, lines 21 to 26: the lack of tuning will most probably be more critical for coupled runs than for atmospheric alone models which are constrained by a fixed SST. This should be mentioned."

We completely agree and have mentioned that now in the manuscript (page 5, line 36-38).

"Section 2, Pages 5 and 6: For clarity I would recommend to put the sections on common forcing fields (2.1, 2.1.1, 2.1.2, 2.1.3) together with the description 3.1 of the Tier 1. For other Tiers, reference to Tier 1 is then sufficient."

Thank you for your suggestion. We prefer, however, the current structure and keep the common forcing fields in a separate section. They are common across the Tiers. Putting them in Tier 1 does not improve the clarity of the manuscript according to us. In addition it is important for the common protocol that the common forcing fields are well discussed. This warrants a separate section.

"There are some redundancies between section 4.2 on links with other MIPs, section 6 on applications and section 7 on analysis plan. I understand the need for some redundancies but I have the feeling the order of arguments could be optimised. For example, it would be clearer to first describe the analysis plan, emphasizing at the same time the related MIPs (eg CORDEX, CFMIP, GMMIP) and some applications related to analyses, and then the list of other potential applications. For the interactions with other MIPS, they could be spread among the analyses and the description of the additional experiments. A table could summarise all these interactions."

We prefer to keep a separate section on the connection with DECK and the other MIPs. This was also required by the WGNM of the WRCP. They want for the different MIPs

explicitly the connection. Incorporating them in the analysis plan would according to us dilute this too much and makes the connections less visible.

We agree with your suggestion of re-ordering the analysis plan and the potential applications and have done this. The analysis plan is now section 6 and the additional potential applications of HighResMIP simulations section 7. In addition we have removed issue 1(Extremes), 5(Scale interactions) and 8(Ocean model biases) and transferred the relevant information to the analysis plan. This reduces, according us, significantly the redundancies. In the analysis plan we have now also made clear the links with the different MIPs for the different specific topics.

We decided not to include an extra table as the topics and connections should now be clear from the text.

"Section 5, page 11, on data: In this part, it is not clear whether the plans for the list of variables to be stored is already fixed or not. Please clarify. "

The data and diagnostic plan will be finalized during the boreal summer of 2016. This is added in the text. (Section 5, line 28)

"It would also be good if an order of magnitude of the storage needed could be given."

The approximate numbers of data storage for the different CMIP6 MIP's, including HighResMIP are provided at http://clipc-services.ceda.ac.uk/dreq/tab01_3_3.html. This is now mentioned in the text. (Section 5, line 32)

"What is meant by the "design of CORDEX will be taken into account"?"

We have removed this phrase. Within HighResMIP protocol we do not describe the forcing to CORDEX. This is left to the individual centers.

"What are priorities 1, 2 and 3 ?"

The HighResMIP data request is based on answering the scientific questions we submitted to the CMIP6 panel: http://clipc-services.ceda.ac.uk/dreq/u/HighResMIP.html

We then needed to find a balance between the data needed to answer as many of these questions as possible, against the ability (and willingness) of as many modeling groups as possible to deliver these high volumes of data. This is how we have prioritized the data request – we consider it should be possible for participating groups to produce all of priority 1 data. We have attempted to organize Priority 2 and 3 data in terms of general usefulness and data volume – priority 2 as lower volume and useful for general questions, priority 3 for more specific questions with very high frequency (and possibly shorter period) output. This has been clarified now in the manuscript.

"Section 6: lines 24-25: Tier 3 experiments are also limited by using atmosphere only models."

We have added this remark in the text (page 16, line 20-21).

"Page 18, the data availability part should not be in this section but rather with section 5."

To put the data availability at the end of the manuscript is a requirement of Geoscientific Model Development. (see http://www.geoscientific-model development.net/about/code_and_data_policy.html) We first put it in section 5 as you suggested, but were requested by GMD to transfer it to the end of the article.

"Technical comments:"

"Page 4, line 11: results rather than representation ?"

Done

"Page 4, lines 30-32: should be more explicit on which resolution."

Done

"Page 5, lines 10-11: the list of model names do not correspond to the list of models used in the references: models MIROC, GFDL, SINTEX-F2 whereas in the references SINTEX-F2, GFDL, Hadley, CESM"

This has been corrected

"Page 5, line 34, please mention explicitly RCP8.5."

Done

"Section 2, Page 5, lines 41-42: the use of a delta to the climatological forcing is not clear enough."

This has been now been made more clear.

"Page 6, Figure 1: would be good to add 1, OA and A for Tiers 1, 2 and 3 respectively on the graphs."

Also in response to reviewer 3 we have extended the figure caption. The information of this figure should now be clear.

"Page 7, line 20. It would be good to add here that 3 runs are recommended but not mandatory (rather than page 8 lines 40-43)"

This is a good suggestion. We have changed the text accordingly.

"Page 7, line 36-37: unclear. Is it for detection/attribution ?"

This simply states the properties of the HadISST2 data set and the uses it can potentially be put to – using the multiple ensemble members is not part of the protocol so we could remove the last part of the sentence, but we thought it was useful to be noted.

"Page 8, line 6: what is EN4 ?"

EN4 is version 4 of the Met Office Hadley Centre "EN" series of data sets of global quality controlled ocean temperature and salinity profiles and monthly objective analyses, which covers the period 1900 to present (Good et al., 2013). This is now explained in the manuscript.

"Page 8, lines 34-38: has already been said"

We agree that we have already discussed this before but now in a different context we want to note this again. We have adapted the text to make this more clear.

"Section 3.2.1, Page 9, 3.2.4 is the use of EN4 recommended ? not fully clear"

EN4 is now stated to be the recommended method. This will make the multi-model ensemble consistent, and enable a systematic investigation of model drift from initial conditions which we think would be a valuable new contribution to CMIP.

"Page 12, line 12 the European Copernicus Climate Data Store"

Thank you. Indeed the order of the words was wrong. This has been corrected.

"Page 13, line 22: rather the air quality than the aerosol only effect on health ?"

Agree. We have changed the text accordingly.

"Page 13, line 32: very few models will use eddy-permitting ocean. This is misleading."

We agree that agree that most of the HighResMIP models will use eddy permitting (∼1/4 degree) and not fully eddy-resolving ocean models. We have modified the text accordingly.

"Table 1: give a reference to "Historic" boundary conditons."

The Historic simulations are part of the DECK and are outlined in Eyering et al. 2016. The reference has been added.

"Tier 2 future add coupled."

Done

Table 9.1:

" what is the standard resolution of NorESM ?" 2 degrees "missing information for GFDL" CM3 standard is 200 km

---

## Author Comment (AC5) · 4 Jul 2016

Response to anonymous referee #3

We thank the reviewer for his/her critical remarks and the positive attitude towards the manuscript.

"My only concern is the lack of a coupled experiment that is likely to have a large signal to noise ratio. An example might be a quadrupling of $CO_2$ at some point during the 1950s control. A similar experiment is proposed in CMIP6 (albeit from pre-industrial conditions). The role of high ocean resolution in heat uptake could provide some results that may be difficult to tease out of the tier 2 simulations. The computation costs could be relatively modest. For instance, Kuhlbrodt et al. (2015) showed useful results from a 20-year experiment."

This type of experiment was already proposed and discussed within the HighResMIP community. In our original protocol it was omitted mainly because to limit the number of experiments. In response to your remark we have discussed this option again and we have now explicitly included this experiment as an additional Tier. We agree with you that it has significant added scientific value and that the computational costs are relatively modest. We have added in 3.4 an additional targeted experiment with an abrupt $4 \times CO_2$ increase in the coupled climate model.

"Minor comments"

"Abstract: define the acronym MIP"

Done

"Introduction: A few more acronyms to define, e.g. ITCZ, MJO, QBO."

Done.

"Section 2.1: Give a proper reference to the CMIP6 publication in this same special issue."

The reference Eyring et al. 2016, that describes the DECK simulations in this special issue, has been added.

"Figure 1: Provide a more detailed caption."

We have provided a more detailed caption. We hope that the content of the figure is clear now.

"Section 3.1.1: As noted we do not have high resolution data for the entire historical period."

We have added a sentence mentioning this.

"Are there any issues with blending the pre and post satellite era data? Whatever methods are used to produce 1/4 degree SSTs, the raw observations simply are not

there. For instance the process outlined for producing future (2015-2050) SSTs relies upon the variability being unchanged in a changing climate. Please comment."

It is true that high resolution observations of SST were not available before the satellite era. The ability to produce 0.25 degree fields of actual SST comes from our use of a satellite-era climatology. The analysis of residuals from that climatology (aka anomalies) which we do on a 5-day timescale is performed on a 1 degree grid. On that spatial and temporal scale, that analysis is perfectly achievable with the data available and is done both for pre-satellite data and satellite-era data alike. Information on covariances between anomalies in different locations gained from the satellite (and in situ) data enables the analysis to be done at the same resolution throughout. The ensemble captures the uncertainty in this process and its spread is wider prior to the satellite era. The Rayner et al manuscript (in prep.) and a technical note for the HadISST (Kennedy, J. et al. in prep.) will describe more in detail the process of generating these data for the past.

For the future, of course you are right that it is a large assumption that the variability will not change. However, we have no other satisfactory way of generating the future forcing, and since the ForcedAtmos future simulations are more useful to contrast different model responses to the same future warming (which is not possible in the coupled models), they are in no way a future projection, so this assumption is not so important.

"Section 4.2, page 10, line 34: insert a space somewhere in andreanalysis."

Done

"Section 6, page 12, last line: No need to define ToE as acronym not used."

Done

"Section 6, Ocean model biases: insert the word "coastal" before upwelling as the equatorial upwelling zone bias is often different to this."

Done

"Section 7.1, page 14, 2nd to last sentence: Is there any evidence that sea ice simulations might improve with increasing resolution. If so give a reference. The project will be moving towards the limit of where the continuum hypothesis is reasonable, which may be an issue."

The reviewer correctly points out that the continuum hypothesis used to model sea ice dynamics may become an issue as we move to high ($\sim$1/4°) and very high ($\sim$1/12°) resolutions. However, there is to our knowledge no scientific evidence that the viscous-plastic or elastic-viscous-plastic rheologies are not adapted (numerically or in terms of results) to such high resolutions. Preliminary tests conducted at 1/4 and 1/12° with the NEMO-LIM3 ocean-sea ice model indicate not only stable results, but also realistic heterogeneities and intermittency behaviors in the sea ice cover. HighResMIP will be the perfect testbed to assess whether these increases in resolution have to be conducted in conjunction with development in model physics (rheology in this case), or if the two can be done separately. We have added a remark about this in the manuscript.

"Section 7.1, page 14, last sentence: This doesn't sound quite right. Maybe Differences would be better than Difference."

Done

"Section 7.1, last sentence: replace "such as" with "outlined by" or something similar."

Done

"Section 7.4, page 16, paragraph 2: Too many acronyms reduce the readability of manuscripts. AR seems a bit unnecessary as it replaces just two words and sonly used twice. TC (a bit later on) also seems a bit unnecessary, but as it is used a few more times I could live with it."

The acronym AR for atmospheric rivers has been removed.

---

## Author Comment (AC6) · 4 Jul 2016

Response to SC4 by Malcolm Roberts

"In order to be consistent with the naming conventions for CMIP6, need to change:

highres-LAI to highresSST-LAI

highres-p4k to highresSST-p4k

highres-4co2 to highresSST-4co2

We recommend that the initial condition for the atmosphere and land for 1950 (for the highresSST-present, and perhaps the highres-1950 experiment) come from the ERA-20C reanalysis from January 1950."

The naming is changed according to the CMIP6 conventions. The recommendation for ERA-20C reanalysis from 1950 is added in the manuscript.

---

## Author Comment (AC7) · 4 Jul 2016

Response to SC3 by Klaus Wyser:

"There is an inconsistency in the suggested GHG forcing (concentrations) between the historical and future run. For the historical run it is suggested to use the new data for CMIP6 but for the future run it will be the old RCP8.5 scenario from CMIP5 (Table 1and 2)."

We agree that there is an inconsistency between using for the historical run the new data of CMIP6 and for the future the old RCP8.5 scenario. The motivation for that is the European H2020 project PRIMAVERA. The partners in this project will start their simulations in the boreal summer of 2016 and will deliver their Tier 3 simulations up to 2050 by the end of 2016. The Tier 2 simulations will be finished in the boreal spring

**[GMDD](https://www.geosci-model-dev-discuss.net/)**

of 2017. The new CMIP6 scenario's will be ready only by the end of 2016 according to present estimates, thereby causing a mismatch with the time schedule of the PRIMAVERA runs. Delaying the PRIMAVERA runs is not a viable option because first of the obligations to the European commission and second because the dependencies of other parts (analysis of the runs, model improvements, impact studies) of PRIMAVERA on those runs. We argue that the switch from CMIP6 historical forcing to future RCP8.5 is an acceptable compromise for the PRIMAVERA partners. The difference between the high-end scenario for CMIP6 and RCP8.5 will be limited up to 2050.

We realize, however, that for other centers, this is a sub-optimal solution and that they would prefer to use a high-end SSPx scenario when it is available. We have therefore decided to make the SSPx as the standard for Tier 2 and 3, and included the RCP8.5 option as a targeted experiment (3.4.e) for those centers that have to start earlier. In the description of this targeted experiment (9.3.5), it is stated that if in a joint analysis the SSPx and RCP8.5 ensembles appears to be significantly different, than the RCP8.5 centers are recommended to repeat their simulations with SSPx, which, due to the short integration period of 36 years, should not be prohibitive.

Below we will in detail discuss the mentioned inconsistencies between CMIP6 and RCP8.5 and how they can be handled.

"This inconsistency causes two problems:"

"1) It is more difficult to have a model version that can handle both the old and new foricng datasets at the same time. It would be pretty easy if it would be just a text file that changes when going from CMIP5 to CMIP6 forcing, but this is not the case. For CMIP6 it is recommended that the models use monthly varying zonal distributions or at least separate GHG concentrations in the two hemispheres (see http://www.climate-energy-college.net/search/content/cmip6). For CMIP5 on the other hand only global annual means have been prescribed. Implementing both options for 1-d and 3-d forcing is not impossible yet certainly nothing positive."

[Figure]

This is indeed an issue. For the historical period we recommend the CMIP6 forcing including the monthly varying zonal or hemispheric distributions. For the future period the distribution of the end of the historical period can be used and subsequently rescaled according to the change in the global annual mean as given by CMIP5. This has the advantage that the forcing structure is according to CMIP6 for historical and future period and that when SSPx is available this will only require a change in the content of the input files.

"2) How consistent are the CMIP5 and CMIP6 forcings? I can imagine that there will be a change in GHG concentrations if the new CMIP6 data are used until 2014, and the old data afterwards. We cannot tell because the future GHG emissions aren't available at this stage, but I am pretty sure that the creators of the dataset will check that historical and future scenarios of the CMIP6 datasets match, but I doubt that they check if historical from CMIP6 and future from CMIP5 data sets fit well together. I would suggest that the GHG forcing is kept consistent across the historical and future simulations (as it is to be done in any other MIP). Either we take the CMIP6 GHG forcing for both phases (preferentially), or the old RCP8.5 from CMIP5, but not a blend of the two."

We agree that there will be no perfect match between the historical CMIP6 and future CMIP5 forcings. We are, however, confident that the mismatch will be small, and can be solved with a smooth interpolation. The advantage is that the historical period, which covers 64 years will then be in agreement with the CMIP6 protocol. The much shorter future period of 36 years will deviate, but can more easily be redone if necessary.